# Nonequilibrium dynamics of spontaneous symmetry breaking into a hidden state of charge-density wave

Faran Zhou[1], Joseph Williams[1], Shuaishuai Sun [1], Christos D. Malliakas [2,3], Mercouri G. Kanatzidis[2,3], Alexander F. Kemper [4] & Chong-Yu Ruan [1✉]

Nonequilibrium phase transitions play a pivotal role in broad physical contexts, from condensed matter to cosmology. Tracking the formation of nonequilibrium phases in condensed matter requires a resolution of the long-range cooperativity on ultra-short timescales. Here, we study the spontaneous transformation of a charge-density wave in $CeTe_3$ from a stripe order into a bi-directional state inaccessible thermodynamically but is induced by intense laser pulses. With ≈100 fs resolution coherent electron diffraction, we capture the entire course of this transformation and show self-organization that defines a nonthermal critical point, unveiling the nonequilibrium energy landscape. We discuss the generation of instabilities by a swift interaction quench that changes the system symmetry preference, and the phase ordering dynamics orchestrated over a nonadiabatic timescale to allow new order parameter fluctuations to gain long-range correlations. Remarkably, the subsequent thermalization locks the remnants of the transient order into longer-lived topological defects for more than 2 ns.

[1] Department of Physics and Astronomy, Michigan State University, East Lansing, MI 48824, USA. [2] Department of Chemistry, Northwestern University, Evanston, IL 60208, USA. [3] Materials Science Division, Argonne National Laboratory, Argonne, IL 60439, USA. [4] Department of Physics, North Carolina State University, Raleigh, NC 27695, USA. ✉email: ruanc@msu.edu

The remarkable feature associated with spontaneous symmetry breaking (SSB) is emergent scale-invariant dynamics in approaching a thermal critical point[1]. There have been strong incentives to understand how this self-organization may proceed out of thermal equilibrium[1,2]. Studying the nonequilibrium phase transition introduced via a swift change of the system parameters, also called a quench, is one of the most active areas in nonequilibrium physics, impacting diverse fields from condensed matter[3,4], quantum gases[2,5,6], to cosmology[4,7]. It is widely believed that, after an interaction quench, isolated systems generically approach a thermal state[2]; however, before thermalization can occur a transient nonthermal stationary state may emerge with properties unlike their equilibrium counterparts[8]. Such investigations have been carried out using ultracold atoms[2,5]. Meanwhile, recently ultrafast pump-probe studies made surprising discoveries of light-induced superconductivity[9] and insulator-metal transitions in hidden charge-density wave (CDW) states[10–12], hinting undisclosed routes towards new broken-symmetry ground states through light excitation far from equilibrium.

In this article, we demonstrate a prototypical case of nonequilibrium SSB into a hidden ground state through a laser-assisted nonthermal quench of the system parameters that define competing orders. The system is $CeTe_3$[13] in which the naturally occurring SSB ground state is the stripe-phase $c$-CDW order[14–16]; see Fig. 1a. Upon femtosecond (fs) near-infrared pulse excitation the pre-existing order is transiently suppressed; however, the system develops a new preference of SSB from the stripe order to a

bi-directional state beyond a well-defined critical excitation. Furthermore, this ultrafast phase transition displays key characteristics of nonequilibrium SSB processes[4,5,17]: the spontaneous emergence of soft-mode instabilities of the new CDW order, followed by a slow onset of the phase ordering stage to develop the long-range correlations. Finally, after the system relaxation back to the thermal ground state the remnants of transient orders survive as long-lived topological defects[4,6,7] for more than 2 ns. The dynamics observed here open an intriguing perspective of controlling phase transitions in quantum materials far from equilibrium.

The $CeTe_3$ studied here belongs to the rare-earth tritelluride ($RTe_3$) family[13,14,16] where the incommensurate CDW develops inside the double Te square lattice sheets, which are isolated by the buckled insulating CeTe layer (Fig. 1a). The 2D-layered $RTe_3$ compounds are ideal systems for studying SSB because in the square net of Te layers two competing stripe density waves (along the $a$ or $c$-axis) can appear[14–16,18]. The formation of different types of CDWs is subject to the nesting in the electronic structure[16,18] but the strong momentum-dependent electron-phonon coupling (EPC) is believed to be essential to form single-wavevector CDW at dimensionality higher than one[19–22]. The shape of the 2D metallic Fermi surface (FS) depends on the relative coupling strengths between neighboring $5p_x$ and $5p_z$ orbitals ($t_\perp$ and $t_{//}$ in Fig. 1a), which play a key role in these interactions[16,18,23,24]. Nonetheless, the tendency for the $RTe_3$ system to form $c$-CDW over $a$-CDW as the preferred broken-symmetry ground state is facilitated by a subtle bi-layer coupling that weakly breaks the C4 symmetry in the Te lattice[18]. Especially in

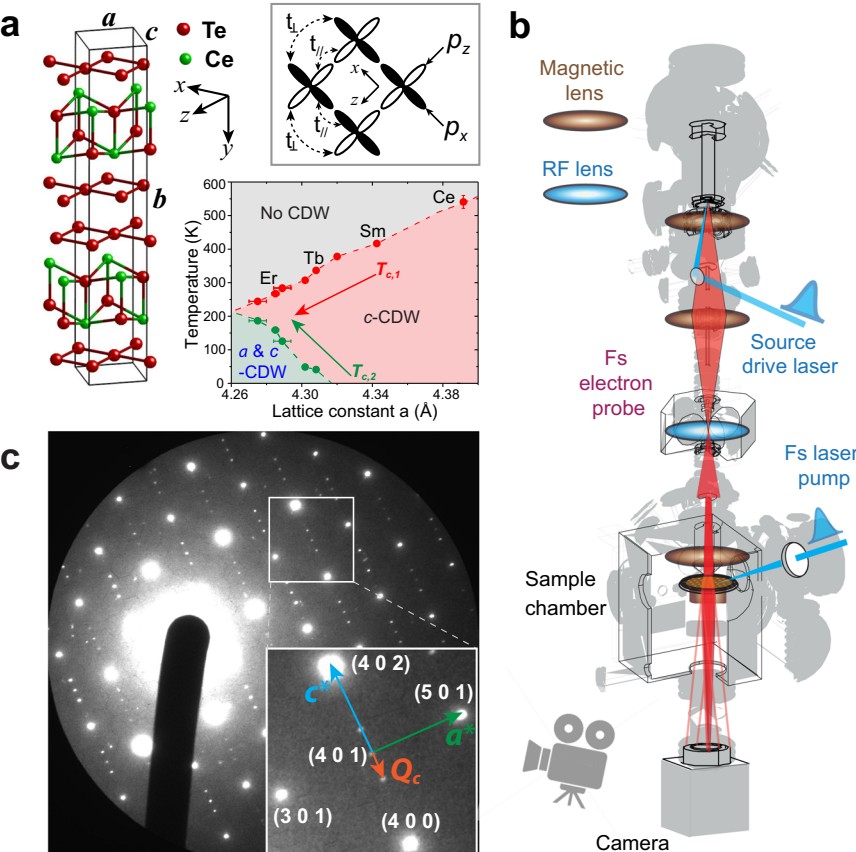

**Fig. 1 Spontaneous symmetry breaking in CeTe3. a** Structure of $CeTe_3$. Corrugated CeTe layers are sandwiched by square Te nets, in which CDW forms. Inside box depicts the Te $5p_x$ and $5p_z$ orbital couplings near the Fermi surface in a unit cell. Inset shows the $RTe_3$ phase diagram with two types of SSB[14,15]. The error bar of CeTe3 is from our TEM measurement while the error bars of other compounds are from the references. **b** Ultrafast electron microscope column with the pump-probe setup for studying the CDW phase transitions. **c** Diffraction pattern of $CeTe_3$ obtained with the fs coherent electron beamline. Inset shows the CDW satellite peaks located at wavevector $\mathbf{Q_c}$ away from the main lattice Bragg peaks in Miller indices. Here $\mathbf{a^*}$ and $\mathbf{c^*}$ are the unit cell vectors of the reciprocal lattice.

CeTe$_3$, a large $c$-CDW gap removes a significant amount of the potential $a$-CDW spectral weight[16,25], which completely excludes subsequent formation of $a$-CDW in its equilibrium phase diagram (Fig. 1a)[15]. Only in the heavier members of RTe$_3$ (see Fig. 1a) where the smaller lattice constant weakens the $c$-CDW can the $a$-CDW develop at a second lower critical temperature[14–16].

Our CeTe$_3$ sample is a single crystal prepared by tape exfoliation to a thickness of ≈25 nm[26] with a transverse size ≈30 μm; see "Methods" section. The sample film thickness is matched to the penetration depth of the pump laser pulse[27,28], establishing a nearly homogeneous excitation profile facilitated by the thin-film interference effect (Supplementary Fig. 1)[29]. The uniformity and thickness of the exfoliated samples are checked using a TEM, where we also conduct the temperature-dependent studies; see Supplementary Fig. 2 for the results. The sample is gently placed on a thin TEM grid in the vacuum specimen chamber where a relatively large pump laser beam (500 μm) illuminates the sample uniformly. Due to the good thermal isolation in this sample setting, the experiments show no visible dissipation of the pump energy absorbed into the materials over the observation window (2 ns). To optimize the probe beam brightness, we deliver intense electron pulses (~10$^6$ e/pulse) at 100 keV generated from a silver photo-cathode just below the virtual cathode limit[30,31]. The phase space of the space-charge-dominated pulse is actively manipulated through placing a radio-frequency cavity in the optical column acting as the longitudinal lens[30,32] (Fig. 1b) to reach ≈100 fs pulse duration. This is accomplished under a condition that its transverse phase space, controlled through a series of magnetic lenses, produces a highly collimated beam[33] with a beam coherence length up to 40 nm[30]. A high-quality pattern produced by this fs coherent scattering setup is shown in Fig. 1c, where the incommensurate CDW state evidenced by the satellite peaks at $\mathbf{Q_c} \approx 2/7\mathbf{c}^{*}$[13,14] stands sharply apart from the lattice Bragg peaks ($\mathbf{G}$) in Miller indices. Due to the large intensity difference between the two (~100:1), typically the lattice peaks are intentionally saturated to provide sufficient dynamical range for investigating the nonequilibrium CDW dynamics.

Dynamics of CDW phase transition in RTe$_3$ have been investigated using the ultrafast pump-probe techniques[24,25,28,34–40]. The pump fs laser pulses couple to the CDW materials with a broad excitation spectrum across the gap, leading to rapid carrier heating. Hence, the natural setting of these experiments has been to investigate the melting of the pre-existing $c$-CDW in this system. The time- and angle-resolved photoemission spectroscopy (trARPES) found that within ≈250 fs upon applying laser pulses, the spectroscopic gap at the Brillouin zone momentum $\mathbf{c}^{*} - \mathbf{Q_c}$ is smeared[24,25,37–39]. The gap dynamics is coupled to the amplitude suppression of the density wave at the momentum wavevector $\mathbf{Q_c}$ as resolved with the fs electron[36,41] and x-ray scattering[40,42]. Time-domain spectroscopies gain accesses to the lower frequency collective modes, which are coherently excited upon applying a weaker laser pulse[20,28,40,43,44]. The BCS-type behavior[14,23,45] has been identified by monitoring the collective mode softening in temperature ($T$) and the laser-fluence ($F$) dependent investigations near $T_{c,1}$[20,43,44]. Focused on the dynamics of the recovery, a study joining ultrafast electron diffraction (UED), optical reflectivity, and trARPES to investigate LaTe$_3$ reported a much faster recovery of gap size than in recuperating the diffraction correlation length. The authors attributed the phenomenon to the presence of topological defects induced in the processes[38]. Interestingly, a more recent MeV-UED investigation following the work found the emergence of a light-induced $a$-CDW state and interpreted the phenomenon as a transient order developed by nucleating around the topological defects[41]. Whereas the various pump-probe investigations have illustrated the ultrafast dynamics of melting and the possibility of defect-induced state in RTe$_3$, the symmetry-breaking aspect that underpins the formation of the hidden order and the associated nonequilibrium phase ordering dynamics have not been clearly elaborated.

Here, with a high momentum resolution fs scattering probe, our work has addressed the nonequilibrium structural formation of a hidden $a$-CDW in CeTe$_3$. The slow onset of the new phase and the coupled soft modes directly involved in the ordering processes lead us to conclude that a nonequilibrium SSB is responsible. The new understanding also bears on a recent clear demonstration of hidden co-instabilities occurring along both symmetry-breaking axes approaching $T_{c,1}$ in a similar DyTe$_3$ system, probed by the inelastic x-ray scattering (IXS)[23]. This result suggests that the underlying energy landscape supports two stripe-phase CDW ground states – even though $a$-CDW remains unsubstantiated throughout the phase transition. The reversal of the symmetry-breaking preference by light excitations evidenced here hence opens a window of opportunities for controlled investigation of the competitively ordered systems.

A phenomenological Landau–Ginzburg (L–G) theory is adopted to capture the essences of nonequilibrium phase transitions. We describe the two CDW states in terms of the order parameter $\eta_l = |\eta_l| e^{i\phi_l}$, where $l = a$ or $c$, and $|(\eta_l)|$ and $\phi_l$ represent the amplitude and phase fields. The L–G free energy modeled on the amplitude field is written as[18,46]

$$f(|\eta_c|, |\eta_a|) = \sum_l \left( \frac{1}{2}\alpha_l \left(T^{(l)} - T_c\right)|\eta_l|^2 + \frac{1}{4}A_l|\eta_l|^4 \right) + \frac{1}{2}\widetilde{A}|\eta_c|^2|\eta_a|^2,$$

(1)

with $\alpha_l$ and $A_l > 0$ for supporting the two symmetry-breaking states at $T^{(l)} < T_c$. The inclusion of the repulsive bi-quadratic coupling term where $\tilde{A} > 0$ makes the eventual symmetry breaking a competition between the two. In this regard, an anisotropy in $\alpha_l/A_l$ makes $c$-CDW instead of $a$-CDW the global ground state. The competing strengths of the order parameters can be evaluated at the stationary conditions, $\frac{\partial f}{\partial |\eta_a|} = \frac{\partial f}{\partial |\eta_c|} = 0$, which define the coordinates of the free energy minimum where $|\eta_a| = \sqrt{\frac{\alpha_a(T_c - T) - \widetilde{A}|\eta_c|^2}{A_a}}$. From this, it is established that a stationary state with a non-zero $|\eta_a|$ occurs only if $|\eta_c|$ is suppressed beyond a critical threshold $|\eta_{c,th}| = \sqrt{\frac{\alpha_a(T_c - T)}{\widetilde{A}}}$, which can occur under the nonequilibrium condition.

In Fig. 2a this nonequilibrium route of phase transition is illustrated. The key step, modeled after the scenario of $F = 1.85$ mJ/cm$^2$, is the sudden unfolding of the L–G landscape from the uniaxial to the bi-directional one, driven by an interaction quench that shifts the pre-existing order parameter $|\eta_a|$ from 1 to ≈$\sqrt{0.17}$ (see stage II). Here, governing the $a$-CDW formation following this change is the projected free energy, $f(|\eta_a|)$, as depicted in the inset panel. The $f(|\eta_a|)$ unfolds from an uphill into a double well under the presupposition that the subsystem effective temperature, $T^{(a)}$, remains near that of the ambient, in contrast to the excited $c$-CDW state. This non-ergodicity presupposition is justified by the high quench speed and the involvement of slow modes mediating the phase transitions. The nonequilibrium scenario makes the bi-directional ordering possible as opposed to the equilibrium route (also shown on the right of Fig. 2a). For details of parameterizing the L–G equation to describe the thermal and nonthermal phase transitions, see "Methods" section.

One chief goal in the studies of nonequilibrium SSB dynamics is to see how the rapidly quenched system adopts the long-range correlations from the deeply unstable order parameter fields[3–5,7,17], often referred to as coarsening[1,3]. While at the late stage of coarsening the system is expected to follow universal laws[1,3], at the

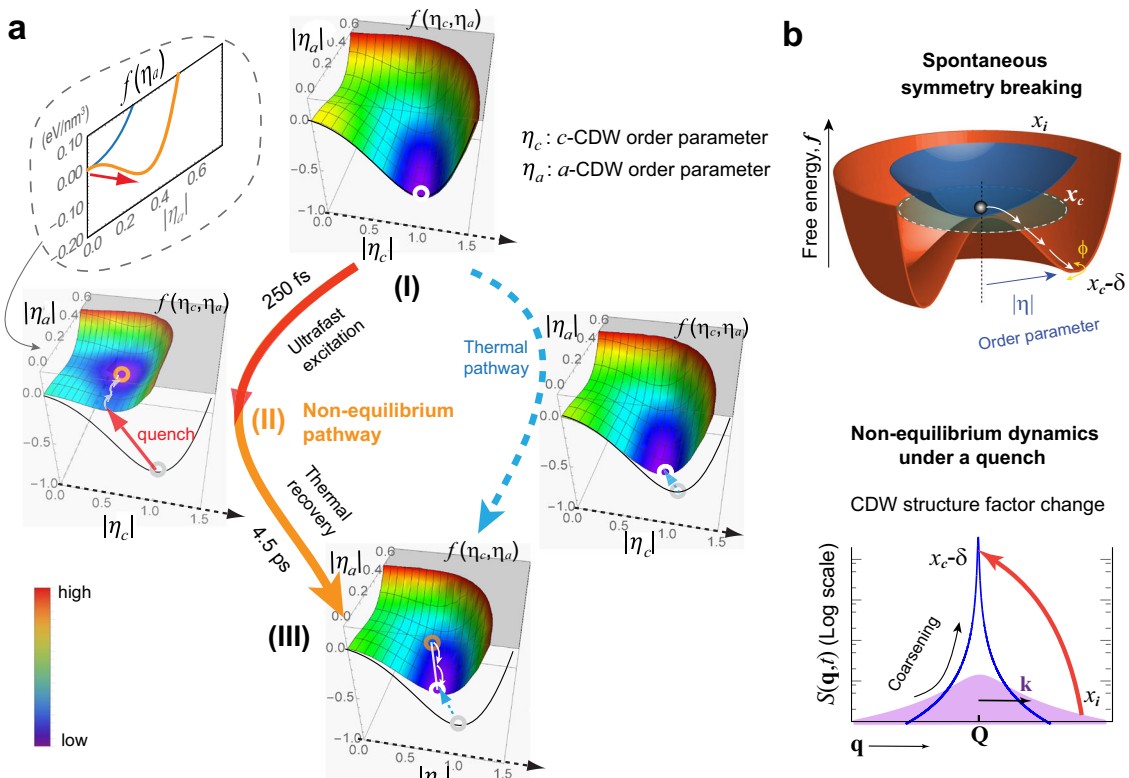

**Fig. 2 Landau–Ginzburg model for nonthermal and thermal CDW phase transitions. a**, The two-dimensional free-energy landscape for photoinduced phase transition modeled after the experiment condition at $F = 1.85$ mJ/cm$^2$. The nonequilibrium pathway (depicted in the left route) involves the formation of a bi-directional hidden state when the repulsive coupling potential is suppressed by a swift reduction of the pre-eminent order parameter $|\eta_c|$. In contrast, in the thermal pathway (right) the global free energy minimum remains in the condition where $|\eta_a| = 0$, namely the CDW is uniaxial. **b** Symmetry-breaking process probed using ultrafast electron scattering. The upper panel shows the free energy unfolding in the order parameter fields of the charge-density wave involving amplitude ($|\eta|$) and phase ($\phi$) degrees of freedom. In a thermal process, the order parameter amplitude grows in response always to the unfolding landscape across the critical point $x_c$. The lower panel shows the scattering profile changes after a rapid quench across $x_c$.

shortest time the evolution is dominated by the spinodal (unfolding) instabilities[17,47]. In CDW phase transitions, the instabilities are mediated by soft modes, which occur in both the amplitude and phase fields[23,48–50], and the order parameter dynamics are governed by a 2D *Mexican hat* free energy (see Fig. 2b) in which the phase fluctuation acts as a Goldstone mode with essentially a zero energy cost[48–51]. It has been theorized that to excite a collective mode in the CDW fields at the momentum wavevector $\mathbf{k} \equiv \mathbf{q} - \mathbf{Q}_a$, two lattice soft phonons at $\mathbf{q} = \mathbf{k} \pm \mathbf{Q}_a$ must be joined coherently[49].

The static single-wavevector lattice distortion wave (LDW), $\mathbf{u}_\eta(\mathbf{Q}_l\ \mathbf{r})$, is a distinct physical manifestation of the CDW state which can be probed directly with the scattering techniques. The conversion between the order parameters (with $|\eta_c| = 1$, $t < 0$) modeled by the L–G theory and the LDW can be established with coefficient $A_{\eta u} = \frac{|u_\eta|}{|\eta|}$. The nonequilibrium LDW with field variations is described by $\mathbf{u}_\eta(\mathbf{r}, t) = u_{0,\eta}\hat{\mathbf{e}}_\eta\left(1 + \delta u_\eta(\mathbf{r}, t)\right)$ $\sin[\mathbf{Q} \cdot \mathbf{r} + \delta\phi(\mathbf{r}, t)]$, where $u_{0,\eta}$ and $\hat{\mathbf{e}}_\eta$ are the amplitude and polarization vector. These field variations, which can be expanded in series of fluctuation waves of wavevector $\mathbf{k}$, manifest in the diffuse scattering centered at $\mathbf{Q}_1$; see Supplementary Note 1. The sum rules allow a simple expression $S_{Q_l}(\mathbf{k}, t) \approx |u_{0,\eta_l}(t)|^2 f(\mathbf{k}, t)$ in which $f(\mathbf{k}; t)$ is a normalized function containing the spatial and ensemble averaging of a system much greater than the CDW correlation length $\xi$; see Supplementary Note 2. In this expression, the structure factor $S_{Ql}(\mathbf{k};t)$ simply is the Fourier counterpart of the autocorrelation function, $S_{\eta_l}(\mathbf{r}, t) \approx \mathbf{u}_{\eta_l}(\mathbf{r}', t)\mathbf{u}_{\eta_l}(\mathbf{r}'', t)$, with

$\mathbf{r} = \mathbf{r}' - \mathbf{r}''$ and the bracket denoting the averaging[1,3,17,48]. The coarsening process, during which fluctuation waves condensed into the LDW, is evidenced in the dynamical structure factor that sharpens over time, as illustrated in Fig. 2b. It is easy to see the amplitude of LDW is given by integrating structure factor over the momenta, $m_{Q_i} = \int S_{Q_i}(\mathbf{k}, t)d\mathbf{k} \approx |u_{0,\eta_l}(t)|^2$. Meanwhile, to probe the fluctuation effects, it is key to resolving $S_{Q_l}(\mathbf{k}, t)$, typically a Lorentzian with $f(\mathbf{k}) \sim \frac{1}{1+\xi^2 k^2}$ (assuming the correlation decays exponentially over the length $\xi$), in the momentum space.

## Results

**CDW evolution after pump/quench.** In Fig. 3, we show the dynamics of nonequilibrium SSB, with the scattering to probe the main event sequences phenomenologically captured by the L–G model. The fluctuation effects associated with the symmetry breaking, which are beyond the mean-field description, are addressed by simultaneously following the scattering structure factors of the main and CDW super-lattice(s). We first focus on the long-range ordering manifested in the LDW. The corresponding structure factors of the bi-directional state, $S_{Qc}(\mathbf{k}, t)$ and $S_{Qa}(\mathbf{k}, t)$, are extracted from the intensity plots over the two satellites across the ordering axes c* and a*, shown in Fig. 3a. Here, the structure factors are obtained in the region near the $G_{401}$ (inset pattern); however, we also confirm the results obtained in other regions are similar. Concerning the immediate impact of the interaction quench, we examine $S_{Qc}(\mathbf{k}, t)$ at $\mathbf{Q}_c = 0.28c^*$, where the scattering in the long-wavelength limit, i.e., at $\mathbf{k} \approx 0$, best

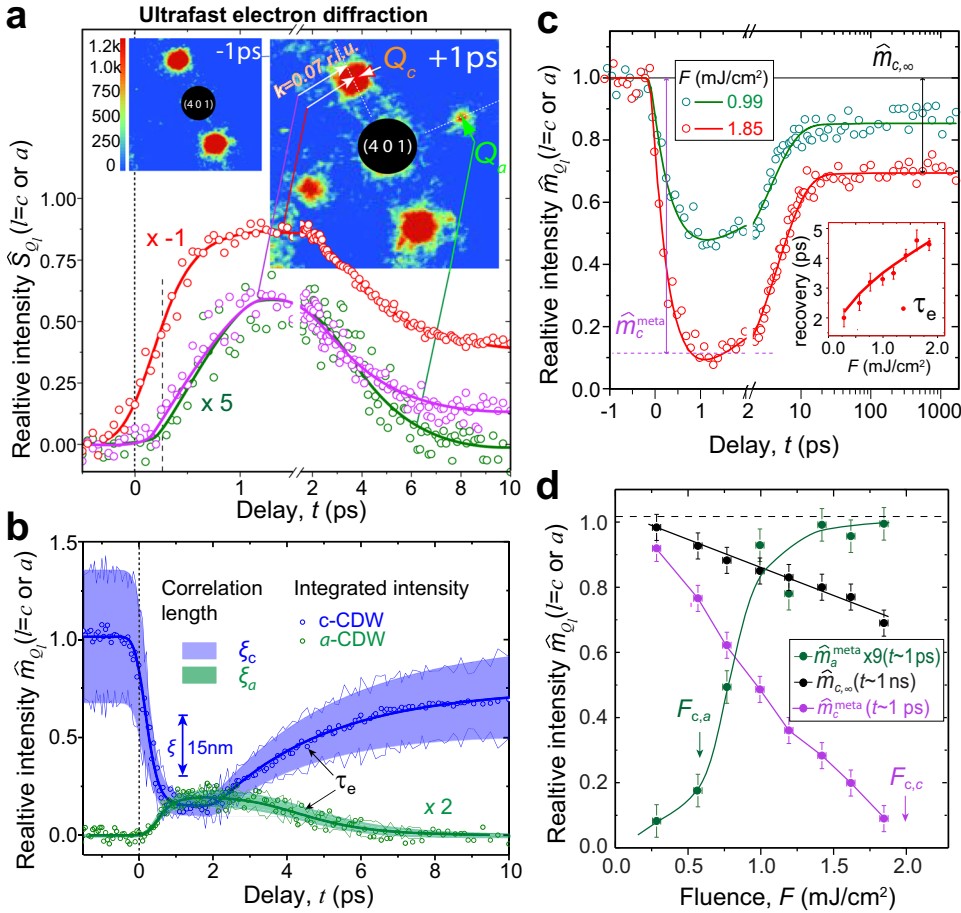

**Fig. 3 Nonequilibrium CDW phase transitions. a** The amplitude dynamics retrieved from the satellite structure factors obtained by ultrafast electron diffraction under $F = 1.85$ mJ/cm$^2$. The dynamics from each CDW order are color-coded: the pre-existing $c$-CDW state in red, and the emerging $c^+$-CDW and $a$-CDW states in purple and green, respectively. Top inset images show the diffraction patterns before and after laser quench with clear emergence of $a$-CDW satellite at $\mathbf{Q}_a$ while the pre-eminent $c$-CDW satellite loses strength. **b** The panel shows the comparison between the evolution of the static $c$-CDW and the emerging $a$-CDW based on integrated intensity. The correlation length ($\xi$) of each state is also presented in the shaded envelop. **c** The intensity evolution obtained for two distinct fluences, showing the general trends of the order parameter evolution with metastable and long-lived stages, where $\hat{m}_c^{meta}$ and $\hat{m}_{c,\infty}$ are extracted. The inset shows the recovery time, characterizing by the relaxation from the metastable to the long-lived stage. The error bars are the standard deviations from the fitting. **d** The order parameter intensities at the metastable (for both $a$ and $c$-CDW states) and long-lived stages extracted from a broad range of fluences. The $F_{c,a}$ and $F_{c,c}$ are determined to be the nonthermal critical thresholds for $a$-CDW formation and $c$-CDW melting. The error bars represent the peak-to-peak noise of the dynamics at negative time delays.

captures the impacts on the pre-existing long-range state by laser excitations (red curve in Fig. 3a). At $F = 1.85$ mJ/cm$^2$, the ordering is suppressed nearly instantaneously, reflected in the halving of the amplitude within the first 250 fs. To study the emergence of $a$-CDW, we turn to the orthogonal sector where new satellites are spotted at $\pm 0.30\mathbf{a}^*$ ($\mathbf{Q}_a$). The scattering intensity at $\mathbf{k} \approx 0$ shows a visible delay, by $\approx 300$ fs relative to the suppression of $c$-CDW. Interestingly, the sampling at a finite spatial frequency, at $\mathbf{k} = 0.07\mathbf{c}^*$, under the extended envelope of $S_{Q_c}(\mathbf{k}, t)$ shows a different response, apparently mimicking the dynamics at $\mathbf{Q}_a$.

**Stabilization of subdominant order**. The results here can be mapped into the L–G order parameter fields. Following the preceding introduction, we calculate the normalized integrated intensity, $\hat{m}_{Q_l}(t) \equiv \frac{m_{Q_l}(t)}{m_{Q_c}(t<0)} = |\eta_l|^2$, which gives straightforward mapping to the order parameter fields, and with the coefficient $A_{\eta u} = u_{0,\eta c} = 0.15$ Å, set by the initial LDW amplitude[52], the LDW amplitudes are also obtained. In Fig. 3b, we plot two trajectories, $\hat{m}_{Q_c}(t)$ and $\hat{m}_{Q_a}(t)$, in response to the transformed free energy. The delayed onset observed in the diffuse scattering retrieved at $\mathbf{Q}_a$ can

be explained with the L–G picture of a serial process for forming the bi-directional order. Following the initial quench, the slow increase of scattering at $\mathbf{Q}_a$ and $\mathbf{Q}_c + 0.07\mathbf{c}^*$ (Fig. 3a) is a signature of coarsening of the system in a newly created but softened 2D landscape (Stage II in Fig. 2a). The system assumes the minimum of the new energy basin, identified by $\hat{m}_{Q_c}(t)$ and $\hat{m}_{Q_a}(t)$ in Fig. 3b, and becomes metastable at $\approx 1$ ps. The field fluctuations that limit the correlation lengths $\xi_c(t)$ and $\xi_a(t)$ are extracted from fitting structure factors $S_Q$; see Supplementary Note 2. The correlation lengths, $\xi_c(t)$ and $\xi_a(t)$, are presented by the colored regions along with $\hat{m}_{Q_c}(t)$ and $\hat{m}_{Q_a}(t)$ in Fig. 3b. The two order parameters straddle for $\approx 1.5$ ps until the relaxation sets in and that drives the system back to the thermal state of $c$-CDW (stage III). The timescale for this thermal recovery ($\tau_e$; see Fig. 3b) is $\approx 4.5$ ps, after which the order parameter does not evolve for a very long time.

**Nonequilibrium critical thresholds**. We find the three main stages of evolution, namely the metastable state, the recovery, and the long plateau, that are also identified at other fluences. We may assume the system reaches thermal equilibrium at the late plateau

stage. Figure 3c illustrates the common dynamics from two selected fluences; for the dynamics over a broad range of fluences, see Supplementary Fig. 3. We focus on the intensities $\hat{m}_l^{meta}$ and $\hat{m}_{c,\infty}$, obtained at the metastable and thermal stages. The $F$-dependent intensities are depicted in Fig. 3d. The results show a threshold behavior for introducing $a$-CDW at $F_{c,a} = 0.59$ mJ/cm². Furthermore, two different critical thresholds for melting, $F_{c,c} \approx 2.0$ mJ/cm² and $F_{c,T} \approx 6.3$ mJ/cm², are obtained by extrapolating $\hat{m}_{Q_c}$ to zero at the metastable and thermal stages. Here, given no dissipation is observed at the late stationary state, the energy density deposited by the laser pulse is well calibrated by equating the absorbed energy at the thermal threshold ($F_{c,T}$) to the enthalpy increase[53] to reach $T_{c,1} = 540$ K (Supplementary Fig. 1), calculated to be $E_{c,T} = 2.0$ eV nm⁻³ ("Methods" section). Interestingly, the nonthermal critical energy $E_{c,c}$ is only $\approx 1/3$ of $E_{c,T}$, signaling the enthalpy increase is localized to only phonons connected to $c$-CDW; namely, the short-time active modes

accounts ~1/3 of the heat capacity according to the difference between $E_{c,c}$ and $E_{c,T}$. This non-ergodicity is enabled by high efficiency of the coupling to the soft phonons/amplitude modes (AM) upon laser excitations, well documented in a number of pump-probe studies[20,28,40,43,44]. This leaves the rest lattice modes only weakly excited at the metastable stage.

**Nonequilibrium critical dynamics and structure refinement.** In Fig. 4a, we outline how the nonequilibrium phase transformation driven beyond the nonthermal critical threshold is expected to proceed. The scenario is given for a deep quench, $F \gg F_{c,a}$, where $f(|\eta_a|)$ unfolds on a timescale over which the critical state dynamics cannot follow adiabatically. Instead, field instabilities would be introduced and the short-range orders appear randomly across the entire sample and coarsen indpendently[3,4,17]. Statistically, the phase dynamics, described by the autocorrelation function, are resolved through Fourier inversion from the

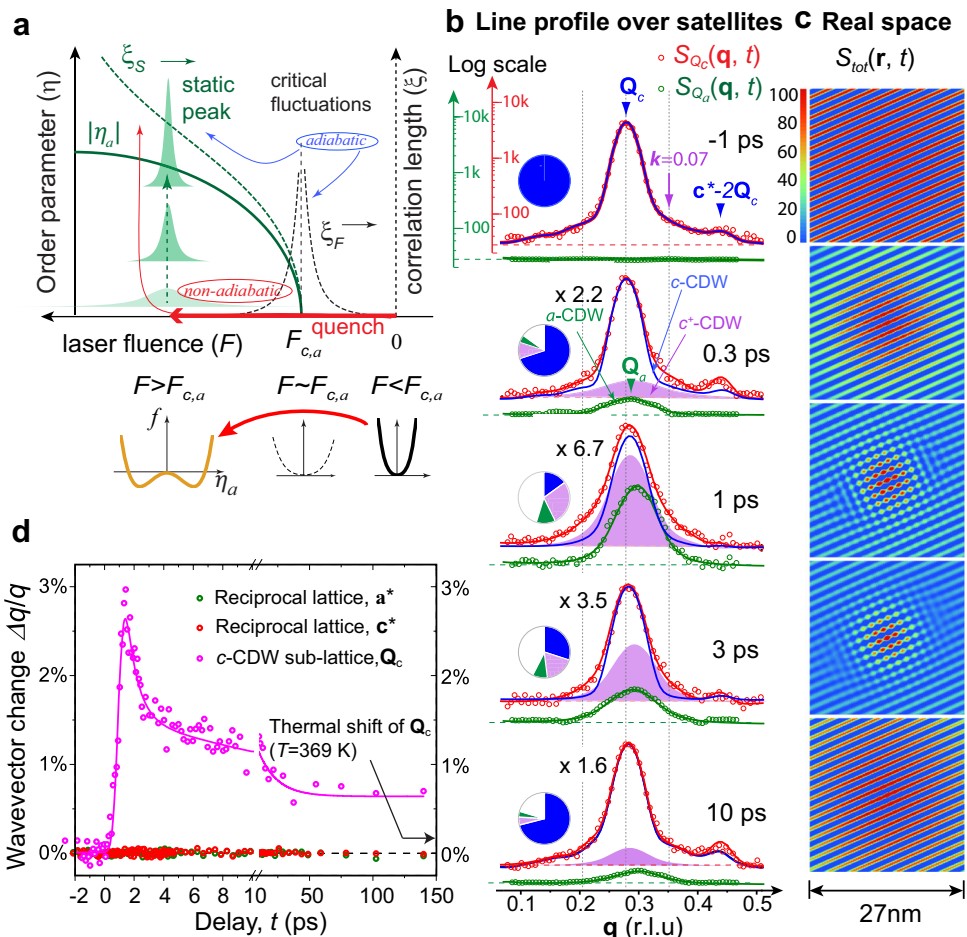

**Fig. 4 Evolutions and refinements of the CDW parameters. a** Schematic depiction of coarsening dynamics (in green) manifested in the structure factor of the $a$-CDW state. The dynamics is driven by an interaction quench of control parameter, here expressed in fluence $F$, across the critical point, $F_{c,a}$. The nonadiabatic growth both in the order parameter amplitude, $|\eta_a|$, and in the growth of the correlation length, $\xi_a$, as reflected in the sharpening of diffraction width, is a different behavior from that of a near-equilibrium route, in which the change in structure factor adiabatically follows the change of control parameter. The critical fluctuations are manifested in the divergence of correlation length, $\xi_F$, near $F_{c,a}$. The free-energy landscapes of the initial state, at the critical point, and the final state are plotted below. **b** Structure factor evolution at $F = 1.85$ mJ/cm², taken along $a$- and $c$-CDW satellites. Note here the semi-log scale to highlight the signals near the diffusive floor where a weak second-order satellite peak at $c^*-2Q_c$ is also identified. The data (in symbols) are fitted with the theoretical structure factor profiles (in line), deduced through Fast Fourier Transform analyses from an interactive structure refinement protocol to construct the autocorrelation function of the system, $S_{tot}(\mathbf{r},t)$, based on the fitting; see Supplementary Note 2. The pie charts (same color codes as in Fig. 3b) indicate the relative scattering weight from each CDW component during evolution. **c** The corresponding $S_{tot}(\mathbf{r}, t)$, deduced from fitting the experimental $S_{Ql}(\mathbf{q}, t)$ in **b**. **d** The momentum wavevector changes obtained for the 2D atomic lattice and the $c$-CDW super-lattice. For the former, the changes are from the lattice Bragg peaks at $\mathbf{Q}_{600}$ and $\mathbf{Q}_{006}$. For the latter, the mean wavevector $\bar{Q}_c$ is calculated by integrating the scattering weight across the $S_{Q_c}(\mathbf{q}, t)$ line profile as shown in **b**.

structure factors (Supplementary Note 2). At this stage of evolution, the diffuse scattering of the collective modes and the structure factor due to the short-range ordering are indistinguishable[3].

An iterative fitting procedure is established, aiming for structure refinement based on a real-space model of LDWs to capture the nonequilibrium evolutions. Based on LDWs, the system autocorrelation function, $S_{tot}(\mathbf{r}, t)$, is calculated to 75 nm, which in turn reproduces the structure factors via a Fast Fourier Transform. The initial input parameters for LDWs, including $Q_l(t)$, $u_{0,\eta_i}(t)$, and $\xi_l(t)$, are deduced from fitting the experimental $S_{Qc}$ and $S_{Qa}$. To deal with the dual order parameter system, with the fluctuations likely subject to the nonlinear effects from the softened modes, a second Voigt function is included to account for the different dynamics shown in the diffuse scattering of $S_{Qc}$ (Fig. 3a). Comparisons between the experimental $S_{Qc}$ (in red dots) and $S_{Qa}$ (in green dots) and the theoretical reconstructions (in solid lines) are shown in Fig. 4b. The theoretical $S_{Qc}$ can reproduce the fine features, such as the high-order peaks at $\mathbf{c}^* - 2\mathbf{Q}_c$, and the diffuse scattering connected with the ordering; for details of the refinement protocol, see Supplementary Note 2. The structure factors derived here are plotted in the semi-log scale in order to adequately depict the evolution of the diffuse scattering. The fitted scattering weight of each CDW state is shown in the color-coded pie chart.

The refinement successfully identifies the second $c$-CDW component, referred hereafter as the $c^+$-component, by its broader scattering feature. The new component (plotted in purple in Fig. 4b) is shown to have a delayed onset and evidently is responsible for the dynamics extracted at $\mathbf{k} = 0.07\mathbf{c}^*$ in Fig. 3a. Another notable feature is the higher ordering wavevector of the $c^+$-component, which is distinguishably greater than the $\mathbf{Q}_c$ of the initial state. In Fig. 4d, we plot the change in the mean position $\bar{Q}_c(t)$ for the entire course of the evolution. The $\bar{Q}_c(t)$ shows similar dynamical features found in the intensity evolution of $a$-CDW. These results, taken together, support that the $c^+$-component is part of the 2D-ordered state, predicted by the L–G theory. We note that the sizeable shift in ordering vectors, from the ≈0.28$\mathbf{c}^*$ to ≈0.3–0.32$\mathbf{c}^*$, has in fact been identified by ARPES in the heavier RTe$_3$ members upon their transitions into the bi-directional states[16,54]. The empirical results show the $a$-CDW would not appear unless the $c$-CDW is weakened. This phenomenon has been attributed to a competition for the spectral weight in the nested FS; the shift in nesting vector is the result of a reduction of FS curvatures[16,54]. Here, the reordering of the $c$-CDW state is triggered by the introduction of the $a$-CDW ordering. Dynamically, the reorganization unfolds in a highly softened landscape in which the modes are anharmonic. The field fluctuations along $|\eta_a|$ can thus effectively couple to the instabilities in the $c$-CDW field, resulting in a co-coarsening. Indeed, during the phase ordering of the $a$-CDW state, the $c^+$ structure factor also shows sharpening, which means the established bi-directional region is expanding over time. This transformation, presented in the autocorrelation function of the system on the length scale of 27 nm, is depicted in Fig. 4c.

**Identification of softened phonons.** Precursors to the symmetry breaking are nonequilibrium lattice excitations in response to the laser quench. The strongest responses occur in modes well-connected to symmetry breaking[22,23]; in particular, in the continuous phase transition[55] as is the case here, the soft phonons are directly responsible for the critical instabilities observed in the dynamical phase transitions[20,43,44]. Unlike the long-range parameters that have been retrieved from analyzing the satellite peaks,

the dynamics of incoherent phonons (soft modes included) are encoded in the intensity decay of lattice Bragg peaks, namely the Debye–Waller factor (DWF)[56], $e^{-2M_{hkl}(t)}$. Here, the $2M_{hkl}$, which is the summed lattice mean-squared ($ms$) vibrations projected along $\mathbf{G}_{hkl}$[56,57], could inform the lattice kinetic energies—equipartitioned among all the modes at equilibrium. However, immediately upon laser excitations, the partitioning of the kinetic energies is necessarily non-ergodic. The lattice responses here are analyzed via the change of DWF (Supplementary Note 1), which is expected to show highly momentum ($\mathbf{G}_{hkl}$)-dependent features due to the symmetry breaking (and recovery) and the non-ergodicity in the phonon system.

In RTe$_3$, the instabilities from soften phonons approaching the thermal phase transition have been probed by the IXS technique[23]. The relevant phonon dispersion curves[23] are schematically reproduced in Fig. 5a. In the nonequilibrium phase transition, the transformations of the dispersion curves are sudden with the most changes occurring in the soft-mode regions (shaded area)[58,59]. Such dynamics can be analyzed through calculating the $ms$ vibrational profile change $\Delta \bar{u}_{hkl}^2(t) = \Delta 2M_{hkl}(t)/|\mathbf{G}_{hkl}|^2$. We note extracting $\Delta 2M_{hkl}$ is an integral part of the systematic scattering profile analysis, which is detailed in Supplementary Note 1. The scheme builds on the scattering weight transfer between the lattice ($S_G$) and CDW ($S_Q$) structure factors and the sum rules to extract the LDW and the DWF dynamics, as outlined in Fig. 5b. The orthogonality of the two CDW systems makes the analyses relatively straightforward. It is worth noting that to form a single-wavevector CDW, the pairing of soft modes into the collective modes and their coalescence into the static LDW shall occur near the minimum of the dispersion curves[45,58]. The dynamical DWF analysis here hence provides us with the rare opportunity to probe the nonequilibrium and momentum-selective events at the earliest stage of symmetry-breaking process, before the order parameter is even substantiated.

Taking advantage of the high-energy electron probe, which enables a large Ewald sphere to cover a broad range of momentum range, the entire momentum ($\mathbf{G}_{hkl}$)-dependence can be mapped without rotating the crystals. The clean way to retrieve the vibrational angular profile rests with analyzing the Bragg peaks over a circular reciprocal lattice zone at the same radius, $|\mathbf{G}_{hkl}|$ (see Fig. 5b). When the symmetry breaking is instigated by the soft modes of a specific polarization and over a limited wavevector range, the angular profile shall exhibit a $\cos^2\theta$ distribution (also $\cos^{2n}\theta$ where $n > 1$ with the involvement of nonlinear interactions) with $\theta$ the angle between $\mathbf{G}_{hkl}$ and $\hat{\mathbf{e}}_q$.

Utilizing Bragg peaks at $|\mathbf{G}_{hkl}| \sim 6$ r.l.u. as shown in Fig. 5b, we have taken the snapshots of the vibrational excitation profile, $\Delta\bar{u}^2(t)$, plotted in the polar coordinates to evaluate the angular dependence. The excitation profiles bear close resemblances to the $\cos^2\theta$-distribution, which is indicative of the soft modes but centered around both axes of symmetry-breaking. We obtain the vibrational amplitude dynamics along $\hat{\mathbf{e}}_q = [100]$ and $[001]$, i.e., $\Delta\bar{u}_{100}^2(t)$ and $\Delta\bar{u}_{001}^2(t)$, in Fig. 5d. The changes are clearly more pronounced along [100], yet both experience a rapid rise on the timescale close to the period of order parameter quench. We can connect the dynamics along [100] to the process of ultrafast melting of the pre-existing $c$-CDW state, where the un-pairing the amplitude mode into soft phonons from the suppression of LDW drives the large $ms$ vibrations along [100]. Naturally, the increase in the vibration amplitude along [001] is connected to the soft modes excited upon introducing the new broken-symmetry landscape, $f(|\eta_a|)$, precursor to the $a$-CDW formation in the system. This analysis also reaffirms the transverse nature of the critical softening, hence the CDW states, a predication made by the recent joint IXS and density-functional-perturbation theory study[23].

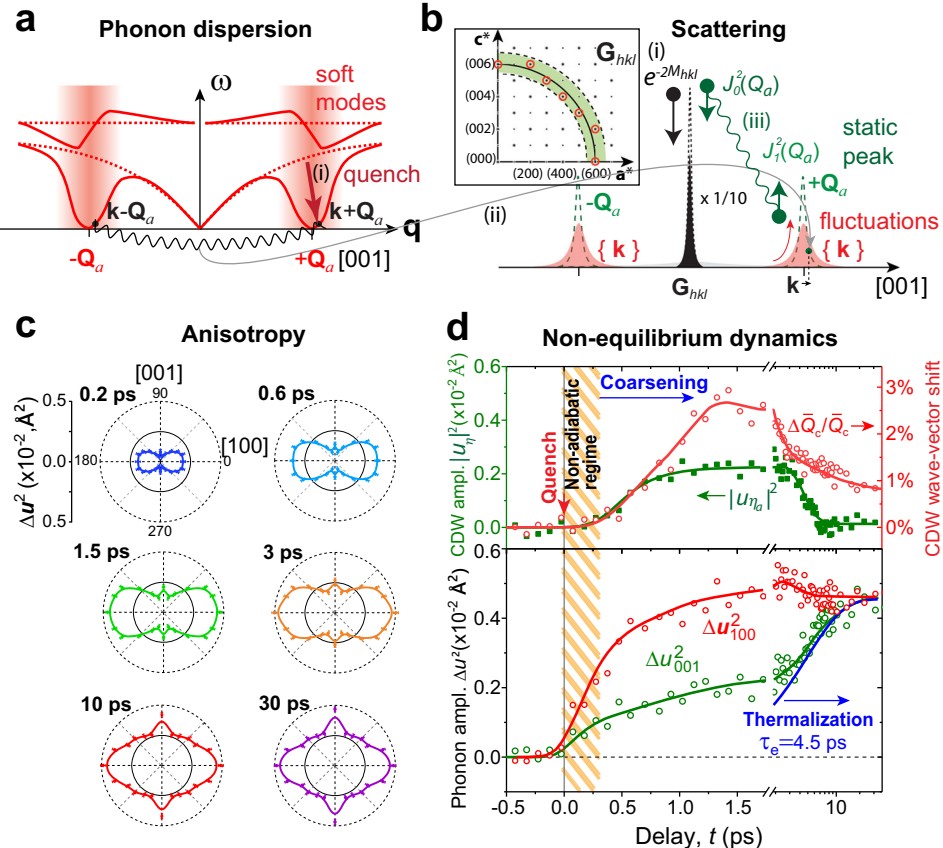

**Fig. 5 Symmetry-breaking and recovery energy landscapes and soft modes dynamics. a** The anticipated change in the phonon dispersion curves associated with a symmetry-breaking phase transition to form the *a*-CDW state. The phonon modes near the wavevector ($Q_a$) of the emerging *a*-CDW is softened (shaded area) upon laser quench. **b** Scattering weight transfer between a lattice Bragg peak ($S_G$) and CDW satellite ($S_Q$) in different stages of symmetry-breaking phase transformation. (1) the excitation of phonons upon laser quench; (2) the transfer of scattering weight from the lattice peak ($S_G$) to the CDW ($S_Q$) peak due to pairing of phonons; (3) the nonequilibrium phase ordering. **c** The mean-squared (*ms*) phonon amplitude changes obtained using Bragg peaks at $|G_{hkl}|$ ~6 r.l.u. (see **b**) shown in polar coordinates, indicating strong anisotropy. **d** (Top) The order parameter field evolution examined via $|u_{0,\eta_a}|^2$ and $\Delta\bar{Q}_c/\bar{Q}_c$, respectively, for *a*- and *c*-CDWs. Here, $\bar{Q}_c$ is the mean wavevector including *c*- and $c^+$ contributions. **d** (Bottom) The vibrational *ms* phonon amplitude changes projected along [001] and [100] obtained from the Debye–Waller analysis.

**Nonthermal phonon manifolds and their equilibration**. The investigation here illustrates the fundamental differences between exciting microscopic soft modes and generating long wave collective modes in density-wave formation. In particular, the time delay between the onsets of soft modes and the order parameter represented by $|u_{0,\eta_a}(t)|^2$ (Fig. 5d), ~300 fs, defines a nonadiabatic time window, which is a distinct signature of the nonequilibrium dynamics near a critical point[4,5]. The two soft-mode branches connected to *c*- and *a*-CDWs are isolated by the vibration polar plots in Fig. 5c. Clearly shown are the modes along the two symmetry-breaking axes having very different amplitude changes; see the bifurcated trajectories of $\Delta u_{001}^2(t)$ and $\Delta u_{100}^2(t)$ in Fig. 5d. Akin to the equilibrium state analyses but considering the non-ergodicity of the transient state, we may use these amplitudes as a measure for the vibrational kinetic energy, or effective temperature, of the subsystems. As discussed, a key to sustaining the hidden state is the non-ergodic environment. The results here thus can substantiate this claim and be used to follow the flow of the kinetic energies in the system. The larger $\Delta u_{100}^2$ indicates the excitation is more readily coupled to the AM of the pre-existing long-range state. The differences between the two are most pronounced in the first 2 ps. Over the longer time, the phonon kinetic energy spreads into the neighboring 2D lattice modes, shown in the reduced anisotropy of the polar plot. We attribute such spreading as an evidence of thermalization between the subsystems. Indeed, this spreading is

linked to a reversal of the bifurcation in $\Delta u_{100}^2(t)$ and $\Delta u_{001}^2(t)$, commencing at ≈2 ps. The changes can be interpreted as a cooling of the *c*-CDW state accompanied by a continuous, but more rapid, heating of the *a*-CDW in the next ~4.5 ps. A direct consequence of this thermal re-equilibration process, as predicted by the L–G model, is the unraveling of the hidden state which is manifested in dynamics of $|u_{0,\eta_a}(t)|$.

## Discussions

We now elaborate on how the new bi-directionally ordered state emerges with a high momentum wavevector and relate it to the reduction in transverse inter-orbital coupling ($t_\perp$ in Fig. 1c) upon laser quench. In RTe₃, the transverse coupling introduces the curvatures in the FS[18]. The curved FS contour leads to different nesting scenarios and allows the anisotropy in the EPC matrix to select a larger nesting vector in the low-symmetry state[19,20,42]. The possibility of symmetry recovery from pulsed laser excitation is described by Rettig et al.[24] in their trARPES experiment, where the FS curvature is reduced in just 250 fs at an absorbed fluence of 0.27 mJ/cm² (equivalent to $F = 0.9$ mJ/cm²). They map this onto a reduction of $t_\perp$ as large as 25% to account the change. Our calculation of the static charge fluctuation susceptibility based on a model band structure (refs. [20,54]) shows that with $t_\perp$ reduced by 25%, the nesting vector $q_{\chi,c}$ shifts towards the lower **q** value by ≈1.25%; see Supplementary Fig. 6. This could explain the increase

of the ordering vector here—since $\mathbf{Q}_c = \mathbf{c}^* - \mathbf{q}_{\chi,c}$[54] and $\approx 2.5\%$ is found at twice the fluence of the trARPES experiment. We point out that our susceptibility calculation does not explicitly consider the formation of $a$-CDW. Nonetheless, empirical data do show the supporting trend that in the heavier RTe$_3$ members (where $a$-CDW is substantiated) the ordering vector of the weakened $c$-CDW shifts to $\approx 0.30$–$0.32\mathbf{c}^*$[16,54]. The movements reflect the reduced FS curvature, hence a more symmetric FS as the system enters the bi-directional state. Given that our observed $\mathbf{Q}_{c+} \approx 0.30\mathbf{c}^*$ and $\mathbf{Q}_a \approx 0.30\mathbf{a}^*$ reflecting the transient FS topology already reach the same level of changes in the bi-directional regime[16,54], it may be concluded that the electronic reorganization induced by the laser excitation may be behind the ultrafast quench mechanism leading to the hidden state.

The long-lived topological defects are widely anticipated in a nonequilibrium SSB process[4,7,17]. Relevant to the ultrafast setting and in the studies of RTe$_3$, several types of defect generating mechanism have been discussed, including localized phase impurities (referred to as topological defects in ref. [38]) from direct high-energy pulse excitations[38,41], the transient domain wall formation driven by nonuniform excitation[28,60], the rapid (re-)emergence of broken symmetry[4,7], and from competing broken-symmetry orders as discussed here. Generally, introducing point defects within the ordered low-symmetry state has a high cost in local energy density, given creating such impurities must shift the phases away from those of the surrounding domains. Furthermore, the presence of defects also softens the CDW sub-lattice, causing amplitude modulation and their impact can be felt far beyond the point defects.

Exploiting the defect-AM interactions, the long-standing annihilation dynamics of topological defects after the nonequilibrium phase transition[4,7,17] were tackled in TbTe$_3$ system with the time-domain reflectivity measurements[28,60,61]. By applying a second weak pump laser pulse to coherently excite the AM in the quenched CDW system, hence allowing measurements of AM damping and linewidth broadening, the authors monitor the annihilation of the topological defects. Interestingly, at least two different timescales were identified. One that occurs in 5–8 ps, which they attribute to the annihilation of the coherent domain walls developed during the inhomogeneous laser quench; a slower process up to $\approx 30$ ps timescale is associated with annihilation of incoherent defects created by Kibble–Zurek mechanism[4,7]; and an even longer timescale of relaxation is inferred[4,7].

Here, the bi-directionally ordered hidden state emerges as a new inhomogeneous component, which play a similar role as the domain wall formation in previous investigations and will impact the ensuing CDW state evolution. The transient inhomogeneities are the sources of Kibble–Zurek-type topological defects created in the (re-)emergence of the broken-symmetry state and the long-term residual effects expected following a rapid cooling[4,7]. We focus on the $\bar{Q}_c(t)$, which is directly sensitive to the reorganization; the dynamics at different stages are featured in Figs. 4d and 5d. The lack of persistent topological defects generations directly from the quench is reflected in the absence of detectable movement in $\bar{Q}_c$ at the early stage (see Fig. 5d; also ref. [42]). The natural type of defects emerging following the formation of hidden states is the dislocation (see Fig. 4c), caused by the strain fields in the CDW super-lattice induced by the local phase modulations—the bi-directional state has a larger wavevector than the uniaxial state. The strain fields and the majority of the topological defects created at the domain boundaries can coherently unwind as the domains of the hidden state recede over the ps thermal recovery process. This is followed by a much slower decay over tens of ps featuring an incoherent process for defect annihilation[60] and some defects could even be frozen out after the system is cooled

sufficiently[4,7]. Indeed, over the ns timescale in which the thermal state is well restored ($\hat{m}_c = 0.7$–$0.8$, consistent with a thermal state at $\approx 369$ K estimated based on the fluence; see Supplementary Fig. 1), an anomalously large shift in $\bar{Q}_c$ of $\approx 0.75\%$ has persisted (Fig. 4d). This shift is more than three times of what expected of the thermal state ($\approx 0.2\%$). The residual effect is a signature that the topological defects created in a rapid nonthermal symmetry-breaking dynamics can be preserved in the system and significantly modify the collective state properties well after the phase transition. Unfortunately, we are unable to monitor the changes beyond 2 ns here although the relaxation on a longer timescale eventually reset the system over the pump-probe cycles ($\leq 1$ ms).

In summary, we have provided high-resolution fs electron scattering data on the hidden state formation in CeTe$_3$, uncovering nonthermal critical onset and bi-directional nature of the light-induced state. Evidence suggests the critical softening to be mediated by the transverse excitation of the phonons that can be traced to the change of the FS topology driven by the transverse inter-orbital coupling suppressed by the fs laser pulses. Perhaps the most important finding here is that, rather than engineering an entirely new metastable landscape, the quench pulses accentuate the intrinsic bi-directional competitive energy landscape that governs both the thermal and nonthermal phase transitions. The nonthermal suppression of the present dominating order tilts the symmetry-breaking preference, thereby introducing the hidden order obscured in the thermal competition. The nonequilibrium physics plays a central role; the system's capability to maintain non-ergodicity during the formation stage allows the new order to flourish and gain long-range correlation. RTe$_3$, while simplest in manifestation, is likely not unique to harbor such hidden states. In light of the phase competition to be a fundamental phenomenon in any high-symmetry, low dimensional systems, it is highly likely that the light-induced hidden state formation would be a common phenomenon in any competitive system. In that case, the out-of-equilibrium approach to accentuate the hidden order and probe the coupling constants could be a powerful approach to explore the fundamental physics often obscured under the equilibrium condition.

## Methods

**Sample and experimental setup.** The single-crystal CeTe$_3$ samples are grown using the chemical vapor transport technique[13] and tape-exfoliated into 30–50 μm sized thin flakes[26] before transferring to a TEM grid as free-standing. Before loaded into the ultrafast beamline, samples were prescreened for uniformity from the contrast in optical microscope and then the thickness ($d = 25 \pm 10$ nm) was determined by performing zero-loss electron energy loss spectroscopy thickness map with TEM. In the pump-probe experiments, the electron beam waist, >100 μm, covers the entire sample. The pump laser beam with S-Polarization is directed at 45° incidence, with the pump fluence estimated by measuring waist radius, $\sigma$, and the pump power $P$: $F = \frac{P}{f_{rep}\pi\sigma^2}$, where $f_{rep} = 1$ kHz is the repetition rate of pump-probe experiment. In Supplementary Fig. 1, the excitation field profile is obtained by solving the Maxwell equations with the boundary condition of the thin film[29] based on the real and imaginary refraction index $n = 1.04$ and $k = 3.45$ determined for CeTe$_3$[27]. It is worth noting that the profile of local absorbed energy density is not an exponential decay, but a modified flat distribution due to the interference effect between the forward and backward fields. Furthermore, the total absorbed energy density by the sample is calculated to be: $E = \frac{FA\cos(45°)}{2\ln(2)d}$, where $A \approx 0.3$ is obtained from solving the Maxwell equations and similarly the transfer matrix method[29,62]. The absorbed fluence is independently verified by measuring the thermal melting threshold as discussed below.

**Landau–Ginzburg free-energy model based on inputs from the experiments.** The normalized intensities $\hat{m}_c^{meta}$ and $\hat{m}_{c,\infty}$ (see Supplementary Fig. 3) across the fluence range are used to evaluate the energy density thresholds for the phase transitions at the thermal and nonthermal regimes; the results are summarized in Fig. 3c. Here, the fluence to the absorbed energy density conversion is well calibrated by the thermal melting threshold at $F_{c,T} \approx 6.3$ mJ/cm$^2$ (Fig. 3d), corresponding to an energy density gain of $E_{c,T}$ 2.0 eV/nm$^3$ needed for heating the sample to $T_c = 540$ K (Supplementary Fig. 2) based on the specific

heat of $\approx 100$ J/(mol·K)[53]. At the nominal fluence of our experiment (1.85 mJ/cm$^2$), the effective temperature $T_{eff}$ of the specimen is $\approx 369$ K (a $\approx 70$ K rise from the temperature of the initial state $T_i = 298$ K). The Landau coefficients can be determined based on the critical energy density scales determined by the experiments. In the L–G model, the critical energy density connected to forming the $c$-CDW state is set by the free energy minimum. Experimentally, supplying an energy density $E_{c,c}$ 0.64 eV/nm$^3$ reduces $|\eta_c|$ from 1 to 0, and from this determination we obtain $\alpha = \frac{4|E_{c,c}|}{T_c - T_i} = 1.05 \times 10^{-2}$ eV nm$^{-3}$ K$^{-1}$, and $A_4 = \alpha(T_c - T_i) = 2.54$ eV nm$^{-3}$.

The remaining Landau coefficients are determined in accordance with

$\alpha' = \frac{|\eta_{c,th}|^2 \widetilde{A}}{T_c - T_i}$, $A'_4 = \frac{\alpha'(T_c - T_i) - \widetilde{A}|\eta_c^{meta}|^2}{|\eta_a^{meta}|^2}$, and $|\eta_c^{meta}|^2 = \frac{\alpha(T_c - T^{(c)}) - \widetilde{A}|\eta_a^{meta}|^2}{A_4}$, where $\widetilde{A}$ is floated as the fitting parameter to fit the experimental parameters, i.e., the $\eta_{c,th} = 0.87$, and the $(|\eta_c^{meta}|^2, |\eta_a^{meta}|^2)$ reported as the coordinates of the free-energy minima as a function of the excitation energy density. We find the data across the fluence range can be best fitted with $\widetilde{A} = 4.10$ eV nm$^{-3}$ and an estimate of $T^{(c)}$ based on involving only 45% of the active specific heat at the metastable stage at $\approx 1$ ps—the more restricted phase space that the early stage excitation couples to and leads to a lower nonthermal melting threshold. Accordingly, we determine $A_4 = 2.54$ eV nm$^{-3}$, $\alpha' = 1.29 \times 10^{-2}$ eV nm$^{-3}$ K$^{-1}$, and $A'_4 = 22.0$ eV nm$^{-3}$. The mean-field energy gain from opening $a$-CDW gap[45] is given by $E_{el} = n(\epsilon_F)\Delta_e^2[1/2 + \ln(2\epsilon_F/\Delta_e)]^2$, where $\Delta_e$ is the CDW gap and $n(\epsilon_F) = 1.48$ state/eV/(u.c.v.) the ungapped density of states near the Fermi energy, $\epsilon_F = 3.25$ eV[16]. The implicit role of EPC lies in the large gap size. Here, the unit cell volume[13] (u.c.v.) is 0.503 nm$^3$. We obtain $E_{el} \sim 1.55$ eV/nm$^3$ based on $\Delta_e = 0.59$ eV[25] and the understanding that 33% of the original FS contour remains metallic in CeTe$_3$[16]. The $E_{el}$ deduced here is similar to the L–G free energy minimum at T $= 0$, i.e., the condensation energy $E_{c,0} \approx 1.63$ eV/nm$^3$.

## Data availability

The datasets generated and analyzed during the current study are available from the corresponding author upon reasonable request.

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

## Acknowledgements

The authors acknowledge help from K. Sun with the TEM measurements, and the helpful contributions from E.A. Nowadnick in the simulation. We also thank M. Maghrebi for insightful discussions. The work at MSU was funded by the US Department of Energy, grant DE-FG0206ER46309. The experimental facility was supported by US National Science Foundation, grant DMR 1625181. A.F.K. acknowledges support from the National Science Foundation under Grant No. DMR-1752713. The work conducted at Argonne National Laboratory was supported by the US Department of Energy, Office of Science, Basic Energy Sciences, Materials Sciences, and Engineering Division.

## Author contributions

C.Y.R., F.Z., and J.W. designed and executed the experiments. C.D.M. and M.G.K. prepared the samples. C.Y.R. developed the software for analyzing the data. F.Z. and C.Y.R. did the data analysis. S.S. did the calculations on the laser excitation profile. A.F.K. did the calculations on the susceptibility and ordering vector.

## Competing interests

The authors declare no competing interests.
