## [Peer Review File · Nature Communications]

Reviewers' comments:

Reviewer #1 (Remarks to the Author):

The manuscript by Zhou et al. "Nonequilibrium dynamics of spontaneous symmetry breaking into a hidden state of charge-density wave", reports on studies of structural dynamics in a charge-density-wave (CDW) system CeTe₃ driven by intense optical excitation and probed by femtosecond electron diffraction. In particular, in CeTe₃ a CDW along the crystalline c-axis is observed below ~ 540 K. Following photoexcitation, the periodic lattice distortion (PLD) associated with the c-axis CDW modulation is strongly suppressed on the sub-ps timescale, and (partially) recovers on the timescale of a few picoseconds. At the same time, a CDW modulation oriented along the a-axis (which is not present under equilibrium conditions) is induced. While the buildup of the a-axis modulation proceeds on a similar timescale as the suppression of the c-axis CDW, there appears to be some time delay between the two processes (from Fig. 3b it also follows that there exists a threshold fluence associated with the appearance of the a-axis modulation). The authors argue, that the dynamics is mediated by soft modes and that "the ultrafast dynamics of the order parameter reflects the inherent broken ergodicity in a spontaneous symmetry-breaking phase transition".

Before addressing details, I should mention the nearly identical results obtained on LaTe₃ (Kogar et al., Nature Physics 2019, <https://doi.org/10.1038/s41567-019-0705-3>, see also arXiv:1904.07472v1). LaTe₃ has nearly identical properties as CeTe₃ (c-axis CDW under equilibrium conditions). The work by Kogar et al., building upon the work of the same group (Reference 31), attributes the appearance of the a-axis CDW order to a competition between the two, assisted by the photoinduced topological defects. Indeed, in rare-earth tritellurides with rear-earth elements with higher atomic numbers (from Tb to Tm) a second CDW transition takes place at low temperatures with additional modulation along the a-axis. Moreover, evidence for photoinduced topological effects was reported in [31].

I am seriously concerned by the lack of mentioning of the work by Kogar et al.. While the paper has been published in November 2019, it has been on arxiv since April 2019! Moreover, even citing [31] seems to be somehow out of context. In my opinion, this is unacceptable.

Given the fact that nearly identical results have been previously published, I do not see this work to match the novelty criteria for publication in Nature Communications. Furthermore, the manuscript does not seem to go beyond the published paper.

In addition, the wording in the manuscript is often confusing, with the line of thought difficult to follow. E.g. the authors claim to "observe a consistent blue shift in the wave-vector..". What does this mean? After reading the MS for several times, I still fail to understand the underlying interpretation, and its differences (if any) with the scenario put forward by Kogar et al. (competing orders). The authors argue to establish the critical fluence $F_{c,a}$ as an (orbital) interaction-mediated critical point, referring to their own (experimental) papers [11,16]. Some elaborated discussion of the proposed scenario should be included!

The authors differentiate between the c-CDW (equilibrium/static CDW order along c-axis) and (c+)-CDW (emerging CDW-order after photoexcitation) – see Figure 3. What is the meaning of (c+)-CDW, given the fact that the corresponding diffraction peak is at the same position, and the original c-CDW peak gets only suppressed in these studies? The two seem to be extracted from the c-CDW peak analysis (Fig. 2c). Given the fact the equilibrium profile (at -1 ps time delay) does not seem to differ much from the profile at the latter time-delays, I am puzzled by this assignment.

The authors use the three-temperature-model to analyze the data. According to these simulations, the temperature of the lattice is increasing on the ~ 5 ps timescale for fluence of 1.86 mJ/cm² (Extended

Data Fig. 5). This timescale seems to be determined from the dynamics of the lattice peaks (Fig. 4b). Since the intensity of the lattice peaks depends on both, the structure factor (i.e. PLD), and random (thermal) displacements, the assignment of the slow 5 ps timescale to thermal motion may not be justified. In fact, the dynamics of the PLD recovery seems to proceed on the ~ 5 ps timescale as well. In the paper by Kogar et al., the authors access the time evolution of the lattice by investigating the time evolution of the inelastic background. The background increases on the ~ 1 ps timescale, with timescale being excitation density dependent (reaching 1.5 ps at fluence of ~ 5 mJ/cm²). Since it is hard to imagine a major difference between the two systems, I suspect that the three-temperature-model analysis need to be revised here.

It is not clear to me, how did the authors obtain the plots in Fig.4a and Fig.4c? Based only on the DW analysis, sketched in Section 7?

In Abstract, the authors state that the transient order (topological defects) last for more than 1 ns. Yet, in all their data the modulation along the a-axis seems to vanish on the 10 ps timescale. What am I missing here?

Finally, looking at the cited works I am missing some additional citations to studies of CDW dynamics using diffraction methods. There are numerous published studies in the last 10 years, using both XRD and electron diffraction. Apart from [31], this paper seems to refer only to publications from the MSU-group.

Reviewer #2 (Remarks to the Author):

The manuscript presents an experimental investigation of the quench and recovery of the CDW order in CeTe₃, using time-resolved electron diffraction. The data, which measure the incommensurate CDW peaks, are interpreted as an intermediate (hidden) CDW phase, which appears during the recovery of the original phase (c-CDW). I read this manuscript with great interest. A detailed study of the ultra-fast spontaneous symmetry breaking dynamics, and the coarsening dynamics during the recovery, in a rather well characterized compound should be of true interest for the study of photo-induced phase transitions in complex materials, which is a large and still growing field. As far as I can see as a theorist, the experimental results are novel and state of the art. The subject matter definitely touches upon fundamental physics.

These are my questions:

(1) Analysis of the scattering peak in Fig 2: It would be good to guide the reader in the main text more directly how the three components (aCDW, cCDW, c*CDW) are extracted. If it is a fit with three convoluted Lorentzians (supplemental) to the lines $I(q)$, one could add an explanatory comment in the main text.

(2) What is the interpretation of the c* phase? If the signature of the c* component is the scattering intensity at the location shown in Fig 2a, it does not really need to be a different phase as compared to c-CDW? Instead, coming close to a symmetry-breaking transition with a softening of the modes, the vibrational modes become more and more anharmonic, which will get mapped to the fluctuations $\langle u^2 \rangle$ and should be expressed in an asymmetric shape of the scattering peaks.

(3) Are aCDW and cCDW different phases? If the cCDW comes from a C2 symmetry breaking, there could be two domains? After a quench of the original order, evidenced by the collapse of the correlation length as shown in Fig 2b, there could be transiently islands with a different orientation. These islands remain small (see correlation length), and eventually dissolve. In the text, it is stated

that "[...] However, our diffraction results clearly indicate a coupled ordering form the two CDWs (a and c+) with a strong possibility of jointly forming a checkerboard state." (page 6), i.e., a different scenario. The manuscript should more clearly discuss how the mixed state is ruled out. Related is the overlaid pattern of the scattering patterns in Fig 2d ... If the relative phase of the a, c, c* contributions is not known, how can one distinguish between an inhomogeneous mixture (with patches of a,c,c*) from a real space order that looks like the one shown?

(4) 3TM: How does the analysis of the relaxation time using the 3TM help to interpret the dynamics. It is said that "The relaxational part of the CDW dynamics starts with the decay of metastable state, which corresponds to the thermalization stage in 3TM". However, both the metastable phase and the normal phase will have a set of CDW phonons, which can be singled out in a 3TM. A 3TM has certainly enough parameters to fit a single fluence-dependent relaxation. Can one be more specific what is learned from the 3TM?

(5) The "critical point": page 8, above Fig 4, it is said that " ... we observe a consistent blue shift in the wave-vector of the hidden state (Fig. 2c) after excitation that agrees entirely with a recovery of the symmetry in inter-orbital coupling both in the trend and the magnitude. This establishes the Fc,a as an (orbital) interaction-mediated critical point". The theory, in turn, shows a crossover in a peak of the bare susceptibility χ , Eq M8 and supplemental Fig 4. A tiny change in the maximum of χ along one direction does not necessarily imply a change of the ordering wave vector (at least χ is just the kernel in the RPA equation that determines the instability), so it is not clear only from supplemental Fig 4 a and b that there would be a critical point. In fact, it is said that there is a *continuous* blue-shift of q_c with V_π . This seems to be consistent with the experimental data, which is nice. If there is a critical point in the theory, it would be good if the supplemental analysis could show it.

In summary, I am intrigued by the nice experimental data, and I believe that this paper could be well suitable for publication in Nature Communications, but before I would recommend the paper I would like to see some discussions on the questions above.

We deeply appreciate the Referees' constructive feedback and comments, which have prompted us to significantly improve the presentation, which have provided a stronger basis to support our conclusions. Below, we present a point-by-point response to the Referees' comments.

Replies to comments by Referee #1:

- 1) The manuscript by Zhou et al. "Nonequilibrium dynamics of spontaneous symmetry breaking into a hidden state of charge-density wave", reports on studies of structural dynamics in a charge-density-wave (CDW) system CeTe₃ driven by intense optical excitation and probed by femtosecond electron diffraction. ... The authors argue, that the dynamics is mediated by soft modes and that "the ultrafast dynamics of the order parameter reflects the inherent broken ergodicity in a spontaneous symmetry-breaking phase transition".

Before addressing details, I should mention the nearly identical results obtained on LaTe₃ (Kogar et al., Nature Physics 2019, <https://doi.org/10.1038/s41567-019-0705-3>, see also arXiv:1904.07472v1). LaTe₃ has nearly identical properties as CeTe₃ (c-axis CDW under equilibrium conditions). The work by Kogar et al., building upon the work of the same group (Reference 31), attributes the appearance of the a-axis CDW order to a competition between the two, assisted by the photoinduced topological defects. Indeed, in rare-earth tritellurides with rear-earth elements with higher atomic numbers (from Tb to Tm) a second CDW transition takes place at low temperatures with additional modulation along the a-axis. Moreover, evidence for photoinduced topological effects was reported in [31].

I am seriously concerned by the lack of mentioning of the work by Kogar et al.. While the paper has been published in November 2019, it has been on arXiv since April 2019! Moreover, even citing [31] (Zhong et al.) seems to be somehow out of context. In my opinion, this is unacceptable. Given the fact that nearly identical results have been previously published, I do not see this work to match the novelty criteria for publication in Nature Communications. Furthermore, the manuscript does not seem to go beyond the published paper.

Our responses: We thank the Referee' thorough review and comments based on recent relevant works.

Indeed, a previous version of the paper by Kogar et al. has been in arXiv (arXiv:1904.07472v1) since April of 2019, which is in fact just 2 days after we submitted our paper to the arXiv (arXiv:1904.07120). This is entirely not coincidental, as both groups were presenting at the same conference and were discussing with participants the new findings. There already were some differences in the interpretation of the results. **While the other group have made the key argument based on the topological defects, following their earlier work in Nature Physics (Zhong et al.), we have argued the case from a rather different perspective.** This is in part due to more evidences emerged in our experiments which made us believe a more proper interpretation should be a spontaneous symmetry breaking (SSB) rather than a defect-initiated process.

We also want to point out that the sample settings in the two experiments are not entirely the same and that demands different experimental approaches. In the work first reported in Nature Physics by Zhong et al. (also in the work of Kogar et al.), the samples were prepared with Ultra-microtomy. Such an approach has been the method of choice for conducting ultrafast electron diffraction experiments, given its ability to produce relatively large samples. In our experiment, instead we used Scotch tape exfoliation. The sample size prepared this way is typically much smaller (30 μm compared to $> 100 \mu\text{m}$ with Ultra-microtomy). While we lost the signals by more than an order of magnitude because of the much smaller sample area, the tape exfoliation is also known to produce homogeneous film with a

lower defect density. We compensate for the loss in signals by employing high-brightness femtosecond electron beams here. As one can see **not only are we able to produce even higher signal strength to study the non-equilibrium dynamics near the diffusive scattering floor, both our time and spatial (in terms of coherence length) resolutions are quite high. This allows us to access the experimental regime not accessible by the other approaches. Such a technical advance in itself is a strong merit of this paper.**

We do agree with the Referee that in terms of identifying a new a -CDW order, in this aspect our paper presents results in CeTe_3 similar to the study of LaTe_3 by Kogar et al.. This is a very important discovery as such a new CDW order was thought to be impossible to form in a thermodynamic environment. It is very important that such a discovery is now being confirmed in two separate systems. That said, we do not agree with the Referee's assessment that our work reported nearly identical results as the paper by Kogar et al. On the contrary, we strongly believe, as illustrated below, there are several key new findings that clearly differentiate our work from the contemporary studies. These key observations evidence that the formation of the hidden state is mediated through a non-equilibrium spontaneous symmetry breaking (SSB) process, involving soft modes. We also discussed evidences of long-lived topological defects as a result of non-equilibrium SSB phase transition. These important observations were not shown in the contemporary studies.

The Referee mentions the observation of topological defects in Ref. [31] by Zhong et al. ([35] in revised manuscript). We want to point out that the term topological defect is used differently here from the mentioned work. According to Ref. [35], the reduction of the correlation length in the c -CDW state was cited as an evidence for topological defects. **Here, we discuss the process from the perspective of non-equilibrium dynamics.** The reduction of correlation length has been one of the signatures for critical fluctuations – especially for a continuous phase transition driven under far from equilibrium condition (namely, under the quenched crossing of a critical point). Such fluctuations are expected to be introduced spontaneously due to instabilities in the CDW system. Therefore, while the defects are formed constantly, they also get destroyed in the processes. We want to point out such transient defects are common in any system undergo order-to-disorder (or normal) phase transition – not entirely restricted to a spontaneous-symmetry-breaking system, which is a main topic of our investigation. Here for clarity reason, we refer to them instead as inhomogeneities due to the transitory nature of such defects and other system (phase and amplitude) instabilities that also reduce the coherence length. However, the term topological defects here (which is largely aligned with the notion by Kibble and Zurek in their seminal works on the defect generations after quenched symmetry-breaking dynamics; see Refs. 4, 7, and 43) is reserved to longer lived defects that form at the boundaries between different coherent domains. Introducing such topological defects is a common feature of a continuous phase transition. As the phase differences cannot be easily reconciled in an SSB system (as this would require changing the phase of an entire domain), **such defects are topologically stable and may become long-lived as discussed here at the late stage.**

Below, we provide key arguments about our major findings supporting the non-equilibrium SSB scenario not covered by the contemporary works.

(i) **Slow (delayed) growth of a -CDW after the rapid destruction of c -CDW.**

As the referee pointed out, the work by Kogar et al., building upon the work of Ref. [35], attributes the appearance of the a -axis CDW order to a competition between the two, assisted by the photoinduced topological defects. In our opinion, if the defects are what mainly responsible for the creation of a -CDW, there should not be a delay between the suppression of c -CDW and the formation of a -CDW. But this is clearly not what we have observed here with our settings. Instead, we show several features observed in distinctive stages that can be directly attributed to a

spontaneous-symmetry-breaking phase transition – see our detailed discussion on page 15 of the revised text. These include the spontaneous emergence of critical fluctuations directly responding to the system quench. Here, because the formation of a new static order is governed by few slow modes near the critical momentum, the dynamics is slowed with an apparently delayed onset of the order parameter. As the system eventually thermally equilibrates, we observe the ps recovery dynamics and the longer-lived topological defects that last for more than 1.5 ns. Therefore, we believe our results illustrate the important role of the non-equilibrium physics that cannot be entirely covered by a competition between the two different CDW orders, assisted by the photoinduced topological defects.

(ii) Emergence of soft modes indicative of SSB.

Our key arguments are based on non-equilibrium SSB. If the formation of *a*-CDW (and *c*-CDW) state is indeed the result of spontaneous symmetry breaking, then one should see soften modes associated with these two branches during phase transitions. Since the *c*-CDW and *a*-CDW have C2 symmetry, therefore their soft mode signatures are easy to identify from looking at the anisotropic signatures of the lattice vibrations. Note, in a recent inelastic x-ray scattering study (Ref. 24) of lattice vibrations [assisted by density-functional-perturbation theory (DFPT) calculation], the soft modes were identified upon approaching the critical point as transverse modes. This is indeed what we have seen in our experiments. By carefully analyzing the anisotropic lattice phonon responses to the laser quench, we are able to directly verify that the soft modes are transverse modes predicted by DFPT. Furthermore, by simultaneously observing the soft mode dynamics and the emergence of an order parameter our experiments provide a unique opportunity for studying the dynamics of spontaneous symmetry breaking. In particular, we can directly link the emergence of the soft modes to the quench of the inter-orbital coupling as identified by the ultrafast ARPES experiments, and a slow growth of the order parameter building upon such soft-mode instabilities in the system. The relevant discussions are now included on pages 14-15 under the sections titled: Critical instabilities: soft modes and Emergences of an order parameter out of equilibrium.

(iii) As to the novelty claim, we have carefully examined the now online version of Nature Physics paper, and did not find any citation to our earlier arXiv paper. We are not aware of the acceptance of this paper at the time of submitting to Nature journal. Our arXiv paper has been circulated and cited in the community since then. We understand publishing in arXiv does not claim the novelty (otherwise our work should be the first reporting the photoinduced a-CDW). Nonetheless, we are now including a note in the end of the paper, noting the recent paper on LaTe₃. We are happy to discuss the differences between the two papers in the main text or in the Supplementary if the Referee sees this as necessary.

We hope with now more detailed deliberation of our results, we may convince the Referee that the scope of our investigation does go beyond the work in comparison, and may be a valuable addition to a rapidly growing field.

2) (a) In addition, the wording in the manuscript is often confusing, with the line of thought difficult to follow. E.g. the authors claim to “observe a consistent blue shift in the wave-vector..”. What does this mean? After reading the MS for several times, I still fail to understand the underlying interpretation, and its differences (if any) with the scenario put forward by Kogar et al. (competing orders).

Our responses: We thank the referee for the important comments. Regarding the key differences between our work and the previous study of LaTe₃, we have addressed it in our reply to Question 1). The wording “blue shift” has been changed to “increase” in our discussion.

(b) The authors argue to establish the critical fluence $F_{c,a}$ as an (orbital) interaction-mediated critical point, referring to their own (experimental) papers [11,16]. Some elaborated discussion of the proposed scenario should be included!

Our responses: Motivated by the Referee’s questions, we have made significant changes in the text to elaborate the meaning of interaction-mediated critical point and nonthermal phase transition based on our experimental findings. The relevant discussions appear on page 11 and 12, under the section discussing nonthermal critical point established via quench of the (inter-orbital) interaction and momentum-dependent electron-phonon coupling. For the convenience of the Referee, we cite the key paragraphs of the revised text below:

“To see how the interaction-related system parameter may be changed by laser excitation, we draw the relevant conclusion from a recent ARPES experiment³⁰ where the curvatures of the FS arcs are suppressed from the reduction of a selective transverse coupling constant²⁷ (t_{\perp} in Fig. 1c). ... This divergence in χ is conveyed to the lattice by the electron-phonon coupling (not modelled here)⁴⁶. Assuming the role of nesting that leads to a static lattice distortion, i.e. phonons softening to zero energy, to form CDW at \mathbf{q}_c , the reduction in $\mathbf{q}_{\chi,c}$ translates to an increase in \mathbf{q}_c ($\mathbf{q}_{\chi,c} = \mathbf{c}^ - \mathbf{q}_c$ ⁴⁷; also see Methods Sec. 5). Our results corroborate with this prediction both in the trend and magnitude. This is demonstrated in Supplementary Fig. 2 and Fig. 4d, Therefore, we are able to establish the direct link between the reduction of t_{\perp} and the increased \mathbf{q}_c , illustrating the important role of nesting even under the non-equilibrium driven scenario. Nonetheless, we should point out that the recent theoretical investigations have also suggested that nesting alone is not enough to explain the formations of CDW in RTe_3 . Instead, the full momentum dependence of both electronic structure and EPC matrix element have to be taken into account^{29,48,49}. To this end, as we will show below, clear signatures of lattice softening with swift anisotropic phonon responses to the photo-excitations have also been recorded in our experiments. **Such momentum-dependent responses representing the soft modes are shown to couple nearly instantaneously to the quench of t_{\perp} demonstrated by Rettig et al., which drives the phase transitions along both directions in our case.**”*

(3) The authors differentiate between the c-CDW (equilibrium/static CDW order along c-axis) and (c+)-CDW (emerging CDW-order after photoexcitation) – see Figure 3. What is the meaning of (c+)-CDW, given the fact that the corresponding diffraction peak is at the same position, and the original c-CDW peak gets only suppressed in these studies? The two seem to be extracted from the c-CDW peak analysis (Fig. 2c). Given the fact the equilibrium profile (at – 1ps time delay) does not seem to differ much from the profile at the latter time-delays, I am puzzled by this assignment.

Our responses: We thank the referee to point this out. Regarding Fig. 2c, the different distributions are plotted in semi-log scale to highlight the changes occurring near the background dominated by the diffusive scattering. Furthermore, different panels are rescaled (noted in the multiplication factor on the left side of the peaks) for better comparison. Thanks to the Referee’s comment, we have made these points clearer in the revised manuscript. The changes are made on **page 8** of the main text, with the rephrased sentences:

*“The scattering profiles recorded along both the [001] and [100] directions let us follow in details the evolving order parameter fields and their instabilities. ... **To observe the momentum-dependent features of the non-equilibrium dynamics near the diffusive floor, the scattering peaks at selected delays are plotted in the semi-log scale and with evolving amplitudes scaled up to a similar level for better comparison (see Fig. 2c).**”*

In addition, we also add clarification sentences in the caption for Fig. 2c (page 7):

“ c , Temporal evolution of the CDW profiles taken along a - and c -CDW satellites. Note here the semi-log scale to highlight the signals near the diffusive floor where a weak 2nd-order satellite peak at $c^ - 2q_c$ is also identified.”*

Regarding how the c^+ -component is obtained, it is extracted from the two-Voigt function fitting of the experimental results outlined in Methods Sec. 3. We want to point out that introducing the c^+ -component as the second Voigt function is necessary to reach a good fitting to the experiments. In the Extended Data Fig. 3, this comparison between 1- and 2-Voigt function fittings of the experiments is made. In addition, we find the increasing c^+ -component to be responsible for the initial increase of the sliced amplitude at $k=0.07 c^*$, which is in contrast to the decreasing amplitude at $k=0$ as shown in Fig. 2a. In particular, the a and c^+ component follow very similar dynamics as shown in Fig. 2a.

Regarding the meaning of c^+ -component, (our special thanks to the Referee 2’s comments) **we have proposed two likely scenarios connecting it to the emergence of the transient order**. This is described in the **1st paragraph on page 9** of the revised manuscript:

“.. the coupled spatiotemporal evolution between the a and c^+ -branches allows us to propose two likely scenarios for the emergences of transient orders. One scenario is that the new CDWs appear outside the diminishing coherent regions of c -CDW. The domains of different orientations may grow independently, or jointly (in this case, it may be considered as a checkerboard) – a real possibility given the similar correlation length observed between the two emerging branches (a and c^+). Alternatively, the a -CDW could appear within the domain of c -CDW. A strongly weakened c -CDW order increases the probability for the competing orthogonal order (now also favored due to changes in the system parameters) to form. Approaching the symmetry-breaking phase transition with softening of the lattice, the modes become increasingly anharmonic and this will get mapped into the critical fluctuations of the c -CDW order parameter and manifested as the c^+ -component.”

- (4) The authors use the three-temperature-model to analyze the data. According to these simulations, the temperature of the lattice is increasing on the ~ 5 ps timescale for fluence of 1.86 mJ/cm² (Extended Data Fig. 5). This timescale seems to be determined from the dynamics of the lattice peaks (Fig. 4b). Since the intensity of the lattice peaks depends on both, the structure factor (i.e. PLD), and random (thermal) displacements, the assignment of the slow 5 ps timescale to thermal motion may not be justified. In fact, the dynamics of the PLD recovery seems to proceed on the ~ 5 ps timescale as well. In the paper by Kogar et al., the authors access the time evolution of the lattice by investigating the time evolution of the inelastic background. The background increases on the ~ 1 ps timescale, with timescale being excitation density dependent (reaching 1.5 ps at fluence of ~ 5 mJ/cm²). Since it is hard to imagine a major difference between the two systems, I suspect that the three-temperature-model analysis need to be revised here.

Our responses: Indeed, there are (at least) two different lattice dynamics components reflected in the earlier studies by the ARPES and diffraction experiments with sub-ps and few ps steps. Indeed, the two different types of lattice responses are also reported here with ≤ 1 ps (strongly-coupled soft modes) and 4.5 ps components as shown in Fig. 4d. We thank the Referee for pointing out the two different time scales also demonstrated in the other work. Such different responses are the reason that one must adopt a three-temperature model (rather than the conventional two-temperature model) to understand the experimental results. We refer to Extended Data Fig. 3 to see the two different dynamics: the faster (~ 1 ps) component is described in the green curve (which represents the phonon branches more strongly coupled to the laser quench, and the slower one (~ 4.5 ps) described by the blue curve represents the general lattice heating. The three-temperature model was initially employed to model such multi-step relaxation processes in RTe₃ reported

Nonetheless, here **the main reason for us to introduce the three-temperature model (3TM) is to understand the recovery time τ_e of the hidden state reported in Fig. 3c, which can vary from 2 to 5 ps strongly depending on the fluence.** It is natural to assume the recovery time is related to the thermalization of the system. In the 3TM framework, the thermalization requires the energy obtained by electrons to be re-equilibrated, and this involves energy transfer from the electrons to a subset of energetic phonons strongly coupled to the electronic perturbation of the system, and also to the rest of the lattice modes that is less efficient. In a simplest form of 3TM, we may write the coupled rate equation $C_i \partial_t T_i = -G_{ij}(T_i - T_j) - G_{ik}(T_i - T_k)$ to describe the energy exchange between the three reservoirs (electrons, strongly coupled phonons and weakly coupled phonons), where G_{ij} is the coupling constant between the reservoirs i and j , and T_i and C_i is the effective temperature and heat capacity for reservoir i . In this picture, the characteristic timescale for energy transfer ($i \rightarrow j$) is given by: $\tau_{ij}^{-1} = G_{ij}/C_i$. Hence, a larger heat capacity typically slows down the dynamics of energy transfer from the reservoir. For considering the electronic relaxation (here $i=e$), the heat capacity $C_i = \gamma T_e$ with γ representing the electronic Grüneisen constant. **It is easy to see that under a higher pumping laser fluence (F), the initial electronic temperature T_e is larger and that leads to a longer relaxation time, namely $\tau_e \propto F$.** In Methods Sec. 6, we show **this simple picture of predicting the fluence dependence survives in a more sophisticated model** that includes the thermal diffusion from the surface to the bulk of the sample for comparing with different types of experiments⁴⁵. **Using the coupling constants reported in Ref. 44, we reproduce the 3TM calculations and demonstrate that indeed the electronic relaxation exhibits a strong dependence on the fluences applied here. This is shown in Fig. 3c where the results of 3TM (solid line) are matched to the data.**

Thanks to Referee's comment, we make our arguments more explicit on **page 11** of the revised text under the section titled: **Thermalization between critical modes and quasi-particle background:**

- (5) It is not clear to me, how did the authors obtain the plots in Fig.4a and Fig.4c? Based only on the DW analysis, sketched in Section 7?

Our responses: We thank the Referee for raising the questions. The curves in Fig. 4a are schematic reproduction of the lattice phonon dispersion curves, reported by Ref. 24. The dispersion curves were obtained using the inelastic x-ray scattering and the density-functional-perturbation theory. To make this more clear, we now explicitly introduce its origin in the 1st paragraph on page 14 of the revised text. As to how we obtain Fig. 4c, indeed it is based on the DW analysis described in Methods Sec. 7. To make it more clear, we have included more details about the analysis in the revised text; see page 14 under the Section titled: **Critical instabilities: soft modes.** For the convenience of the Referee, we cite the key excerpts of the revised text below:

“Here, for investigating the soft modes we track the (normalized) intensity evolution of Bragg reflections $\hat{I}(G_{hkl}, t)$ with $|G_{hkl}| \approx 6$ r.l.u. as depicted in Fig. 4b. The anisotropic responses expected of system symmetry breaking (or recovery) are studied in the angular dependence of such changes... We note that both the soft modes and the PLD, which evolves as the phase transitions proceed, contribute to the anisotropic intensity changes observed in $\hat{I}(G_{hkl}, t)$. As the PLD amplitude (A) is pre-determined via the intensity integral of the satellites (i.e. $A^2 \sim I_{int,CDW}$), we can exclusively resolve the soft modes contribution. This is accomplished by taking the logarithm of $\hat{I}(G_{hkl}, t)$ and subtract it with the PLD-associated changes – a procedure known as the Debye-Waller analysis; for details, see Methods Sec. 7. The results are determined for each projection (along $[hkl]$) in terms of the mean-squared amplitude denoted as Δu_{hkl}^2 from each selected G_{hkl} . We note that the typical thermal lattice excitations are expected to be small (and more isotropic) and occur on a slower (few

ps) timescale. As major changes in the evolution of Δu^2 are observed at sub-ps, they are expected to be dominated by the soft modes excited in the system. This is clearly shown in its highly non-uniform angular distribution in Fig. 4c, where Δu^2 is plotted in the polar coordinates as a function of delay.”

- (6) In Abstract, the authors state that the transient order (topological defects) last for more than 1 ns. Yet, in all their data the modulation along the a-axis seems to vanish on the 10 ps timescale. What am I missing here?

Our responses: We thank the Referee for raising the questions. The answer to this question might have to do with our definition of the topological defect. We use the term topological defects differently from Refs. [35] and [58], which we have discussed in our reply to Question 1). Motivated by the Referee’s question, we have revised the text to clearly define the meaning of topological defects in the context of Kibble-Zurek theory of non-equilibrium phase transition early on. This appears on page 6 of the revised text:

*“After the quench across a critical point, the system acquires a non-zero expectation of the new order parameter, but the phase (ϕ) is arbitrarily chosen in different regions. When such regions meet after coarsening, they must be separated by a domain wall across which ϕ goes from one value to another. Such a wall is topologically stable. **A closed wall bounding a finite domain may shrink and eventually disappear, but this is a relatively slow process and may have lasting effects**⁴³. We refer to such defects as topological defects as often discussed in the context of Kibble-Zurek theory of non-equilibrium phase transition^{3,4,6,16}.”*

We hope with this clarification it will be more obvious for the readers to follow our arguments about why the topological defects are considered as the remnants of the transient order and are long-lived even after the system recovers to the thermal state.

- (7) Finally, looking at the cited works I am missing some additional citations to studies of CDW dynamics using diffraction methods. There are numerous published studies in the last 10 years, using both XRD and electron diffraction. Apart from [31], this paper seems to refer only to publications from the MSU-group.

Our responses: We very much agree with the Referee that introducing the prior ultrafast experiments on different RTe_3 systems are highly important for our discussions. In the revised text, we have added an entire section discussing the prior works and the findings that are most relevant to our work here. Such reviews appear on pages 5-6 of the revised text under the Sections: **Quasi-particle responses to laser excitations and Non-equilibrium critical dynamics.**

Replies to comments by Referee #2:

The manuscript presents an experimental investigation of the quench and recovery of the CDW order in CeTe₃, using time-resolved electron diffraction. The data, which measure the incommensurate CDW peaks, are interpreted as an intermediate (hidden) CDW phase, which appears during the recovery of the original phase (c-CDW). I read this manuscript with great interest. A detailed study of the ultra-fast spontaneous symmetry breaking dynamics, and the coarsening dynamics during the recovery, in a rather well characterized compound should be of true interest for the study of photo-induced phase transitions in complex materials, which is a large and still growing field. As far as I can see as a theorist, the experimental results are novel and state of the art. The subject matter definitely touches upon fundamental physics.

....

In summary, I am intrigued by the nice experimental data, and I believe that this paper could be well suitable for publication in Nature Communications, but before I would recommend the paper I would like to see some discussions on the questions above.

Our response: We thank the Referee's encouraging comments about the novelty and quality of our work. We also thank the Referee for posing insightful comments, which are quite useful for us to improve the text.

- 1) Analysis of the scattering peak in Fig 2: It would be good to guide the reader in the main text more directly how the three components (aCDW, cCDW, c*CDW) are extracted. If it is a fit with three convoluted Lorentzians (supplemental) to the lines I(q), one could add an explanatory comment in the main text.

Our responses: We thank the Referee's questions. Indeed, the c^- and c^+ -components are deduced based on two-Voigt function fitting of the scattering intensity along the [001] direction; whereas the a-CDW component is fit with just 1-Voigt function along the [100] direction. We have followed the Referee's advice to include the discussions about data fitting in the text. This appears on **page 8**:

"The contribution from each CDW branch as represented by the respective structure factor $S_l(\mathbf{q}, t)$ (l denotes the different CDW branches) is obtained here in the multi-component refinement of our experiments; see Methods Sec. 2 for details of the procedure. ... Such analyses allow us to identify a new ordering component developing in the c-CDW field responsible for the intensity increase at $\mathbf{k}=0.07\mathbf{c}^$, which is coupled dynamically to the formation of a-CDW. Its onset is seen at 300 fs where a separate c-CDW component (colored in purple shade and referred to as c^+ -component hereafter) is captured via fitting the peak profile along [001] using the two-Voigt function; **here c^- and c^+ -components are modelled as two separate Voigt functions** – see Methods Sec. 3. In addition, the peak profile along [100] representing a new a-CDW (green open circles) is also plotted."*

- 2) What is the interpretation of the c^* phase? If the signature of the c^* component is the scattering intensity at the location shown in Fig 2a, it does not really need to be a different phase as compared to c-CDW? Instead, coming close to a symmetry-breaking transition with a softening of the modes, the vibrational modes become more and more anharmonic, which will get mapped to the fluctuations and should be expressed in an asymmetric shape of the scattering peaks.

Our responses: We thank the Referee for very insightful comments. Indeed, we believe that the presence of c^+ -component not as an independent channel is a real possibility. In the revised text, we have included such an alternative interpretation of our results in the paper, which appears on page 9:

“Even though the phase between these CDW states are unknown here – so it is not possible to resolve the exact structure, the coupled spatiotemporal evolution between the a and c^+ -branches allows us to propose two likely scenarios for the emergences of transient orders. One scenario is that the new CDWs appear outside the diminishing coherent regions of c -CDW. The domains of different orientations may grow independently, or jointly (in this case, it may be considered as a checkerboard) – a real possibility given the similar correlation length observed between the two emerging branches (a and c^+). Alternatively, the a -CDW could appear within the domain of c -CDW. A strongly weakened c -CDW order increases the probability for the competing orthogonal order (now also favored due to changes in the system parameters) to form. Approaching the symmetry-breaking phase transition with softening of the lattice, the modes become increasingly anharmonic and this will get mapped into the critical fluctuations of the c -CDW order parameter and manifested as the c^+ -component.”

- 3) Are aCDW and cCDW different phases? If the cCDW comes from a C2 symmetry breaking, there could be two domains? After a quench of the original order, evidenced by the collapse of the correlation length as shown in Fig 2b, there could be transiently islands with a different orientation. These islands remain small (see correlation length), and eventually dissolve. In the text, it is stated that “[...] However, our diffraction results clearly indicate a coupled ordering form the two CDWs (a and c^+) with a strong possibility of jointly forming a checkerboard state.” (page 6), i.e., a different scenario. The manuscript should more clearly discuss how the mixed state is ruled out. Related is the overlaid pattern of the scattering patterns in Fig 2d ... If the relative phase of the a , c , c^* contributions is not known, how can one distinguish between an inhomogeneous mixture (with patches of a, c, c^*) from a real space order that looks like the one shown?

Our response: thank the Referee for insightful comments. We believe this question is related to the question 2). We Indeed, we could not rule out the mixed state scenario, and both scenarios (mixed and co-existent) are now presented in the revised text, as discussed in our answer to question 2).

- 4) 3TM: How does the analysis of the relaxation time using the 3TM help to interpret the dynamics. It is said that “The relaxational part of the CDW dynamics starts with the decay of metastable state, which corresponds to the thermalization stage in 3TM”. However, both the metastable phase and the normal phase will have a set of CDW phonons, which can be singled out in a 3TM. A 3TM has certainly enough parameters to fit a single fluence-dependent relaxation. Can one be more specific what is learned from the 3TM?

Our responses: We thank the Referee for raising the important question. Indeed, the three-temperature model is an effective model aiming to capture the multi-step relaxation dynamics of the hot carriers involving different branches of phonon modes. And yes, the process could be even more complicated and can still be treated by such an effective model to some degree. So fitting a specific timescale is not our main aim here. What we really learn from the 3TM is the fluence(F)-dependent recovery time of the hidden state τ_e , as shown in Fig. 3c. It is natural to assume the recovery time is related to the thermalization of the system given the nonthermal character of the hidden state. In the 3TM framework, the thermalization requires the energy obtained by electrons to be re-equilibrated, and this involves energy transfer from the electrons to a subset of energetic phonons strongly coupled to the electronic perturbation of the system, and also to the rest of the lattice modes that is less efficient.

In a simplest form of 3TM, we may write the coupled rate equation $C_i \partial_t T_i = -G_{ij}(T_i - T_j) - G_{ik}(T_i - T_k)$ to describe the energy exchange between the three reservoirs (electrons, strongly coupled phonons and weakly coupled phonons), where G_{ij} is the coupling constant between the reservoirs i and j , and T_i and C_i is the effective temperature and heat capacity for reservoir i . In this

picture, the characteristic timescale for energy transfer ($i \rightarrow j$) is given by: $\tau_{ij}^{-1} = G_{ij}/C_i$. Hence, a larger heat capacity typically slows down the dynamics of energy transfer from the reservoir. For considering the electronic relaxation (here $i=e$), the heat capacity $C_i = \gamma T_e$ with γ representing the electronic Grüneisen constant. **It is easy to see that under a higher pumping laser fluence (F), the initial electronic temperature T_e is larger and that leads to a longer relaxation time, namely $\tau_e \propto F$.** In Methods Sec. 6, we show **this simple picture of predicting the fluence dependence survives in a more sophisticated model** that includes the thermal diffusion from the surface to the bulk of the sample for comparing with different types of experiments⁴⁵. **Using the coupling constants reported in Ref. 44, we reproduce the 3TM calculations and demonstrate that indeed the electronic relaxation exhibits a strong dependence on the fluences applied here. This is shown in Fig. 3c where the results of 3TM (solid line) are matched to the data.**

Thanks to Referee's comment, we now make our arguments more explicit on **page 11** of the revised text under the section titled: **Thermalization between critical modes and quasi-particle background.**

- 5) The "critical point": page 8, above Fig 4, it is said that " ... we observe a consistent blue shift in the wave-vector of the hidden state (Fig. 2c) after excitation that agrees entirely with a recovery of the symmetry in inter-orbital coupling both in the trend and the magnitude. This establishes the Fc,a as an (orbital) interaction-mediated critical point". The theory, in turn, shows a crossover in a peak of the bare susceptibility χ , Eq M8 and supplemental Fig 4. A tiny change in the maximum of χ along one direction does not necessarily imply a change of the ordering wave vector (at least χ is just the kernel in the RPA equation that determines the instability), so it is not clear only from supplemental Fig 4 a and b that there would be a critical point. In fact, it is said that there is a *continuous* blue-shift of q_c with V_{π} . This seems to be consistent with the experimental data, which is nice. If there is a critical point in the theory, it would be good if the supplemental analysis could show it.

Our responses: We thank the Referee for insightful comments. We agree with the Referee that considering the change in the maximum of susceptibility alone is not adequate to decide the establishment of a critical point. Instead, the full momentum dependence of both electronic structure and electron-phonon-coupling (EPC) matrix element have to be taken into account^{29,47,48}. We have to admit that modeling the latter effect is beyond the scope of our current work. Nonetheless, following the Referee's leads, we have revised the manuscript to discuss not just the role of nesting, but also the momentum-dependent electron-phonon coupling for establishing the critical point. The relevant discussions appear on pages 11-12. For the Referee's convenience, we cite the excerpts here:

*"To see how the interaction-related system parameter may be changed by laser excitation, we draw the relevant conclusion from a recent ARPES experiment³⁰ where the curvatures of the FS arcs are suppressed from the reduction of a selective transverse coupling constant²⁷ (t_{\perp} in Fig. 1c). ... **This divergence in χ is conveyed to the lattice by the electron-phonon coupling (not modelled here)**⁴⁶. **Assuming the role of nesting that leads to a static lattice distortion, i.e. phonons softening to zero energy, to form CDW at \mathbf{q}_c , the reduction in $\mathbf{q}_{\chi,c}$ translates to an increase in \mathbf{q}_c ($\mathbf{q}_{\chi,c} = \mathbf{c}^* - \mathbf{q}_c$ ⁴⁷; also see Methods Sec. 5). Our results corroborate with this prediction both in the trend and magnitude. This is demonstrated in Supplementary Fig. 2 and Fig. 4d, Therefore, we are able to establish the direct link between the reduction of t_{\perp} and the increased \mathbf{q}_c , illustrating the important role of nesting even under the non-equilibrium driven scenario. **Nonetheless, we should point out that the recent theoretical investigations have also suggested that nesting alone is not enough to explain the formations of CDW in RTe_3 . Instead, the full momentum dependence of both electronic structure and EPC matrix element have to be taken into account**^{29,48,49}. To this end, as we will show below, clear signatures of lattice softening with swift anisotropic phonon responses to the photo-***

excitations have also been recorded in our experiments. Such momentum-dependent responses representing the soft modes are shown to couple nearly instantaneously to the quench of t_{\perp} demonstrated by Rettig et al., which drives the phase transitions along both directions in our case.”

Reviewers' comments:

Reviewer #1 (Remarks to the Author):

In their rebuttal and the revised version of the manuscript the authors improved both the arguments as well as presentation. However, I still have problems connecting the proposed interpretation with the presented data (see below).

First, I apologize for not realizing that the first version of this publication has been posted on arxiv even before the post (arXiv:1904.07472v1) of a recently published work by Kogar et al. (Nature Physics 2019, <https://doi.org/10.1038/s41567-019-0705-3>). Given the fact that the two works present (in my opinion) nearly the same result, I can understand the authors' frustration when receiving my report.

Apparently, the publication of "Nonequilibrium dynamics of spontaneous symmetry breaking into a hidden state of charge-density wave" has already been declined elsewhere, before submitting it to Nature Communications. Perhaps one of the reasons for declining publication was the "novelty" argument in view of the similar work by Kogar et al. (which would be unfortunate). On the other hand, I can imagine the reason being the mentioned lack of support for the proposed scenario. Moreover, instead on using the published results and the interpretation as a starting point (pointing out the additional information that can be obtained from the data), the authors simply make a remark that similar results have been published, without critical assessing the data and the proposed model. In my opinion, this is contra productive.

I am turning now to the revised manuscript at hand. The work reads well up to the page 8, when the main experimental result is being discussed (in reference to Figure 2 on page 7). Here I get stuck and frustrated. In my opinion, this part presents the main problem of the manuscript. If I understand correctly, the support for the proposed scenario should be given by Figure 2c, showing the time evolution of the diffraction intensity profiles in the vicinity of the CDW wavevectors q_c and q_a . First, it would be easier for the reader to have the profiles along the c^* and a^* directions plotted separately. Moreover, it would be nice to note that traces in Fig. 2a have different signs. However, my main problem lies in the claimed appearance of a new (transient) order along c^* at the wavevector c_+ different from the equilibrium one (q_c). On page 8 it is argued that $q_{c+} = q_c + 0.07 c^*$, i.e. $q_c = 0.28 c^*$, while $q_{c+} = 0.35 c^*$. This follows from the definition of $k = q - q_c$ on page 8, as well as from the arrow in the upper panel of Figure 2c. I simply do not see any evidence supporting this claim in the raw data (neither in the main text, nor in the supplementary). On page 12, however, the authors quote $q_{c+} = 0.29 c^*$, which is more along the lines with the data in Panel 2c and can barely be distinguished from $q_c = 0.28 c^*$. The authors argue that the distinction between the two orders/periodicities (at q_c and q_{c+}) can be made from the analysis of the superlattice peak with two Voigt functions. All I can see here – in the first approximation – is a broadening of the linewidth. But I see evidence in the data for a major change in periodicity along the c -axis direction (i.e. $q_{c+} = 0.35 c^*$). Perhaps a two Voigt function fit is indeed better than a single Voigt function, but this should not be much of a surprise as the system after excitation is likely inhomogeneous. The simple reason could be that the excitation profile as such is inhomogeneous, since the optical penetration depth (20 nm) is comparable to the sample thickness (25 nm). (Note that in the work by Kogar et al. thicker samples are used, making the excitation even more inhomogeneous. This is manifested by weaker suppression of the CDW peak at q_c at higher excitation densities). Anyway, since both referees pointed out the issue of q_{c+} vs. q_c in original reports, I would have expected the authors to address it.

The way I understand it, assuming a new periodicity of the new order with changed periodicity along the c -axis direction is the most important (experimental) distinction of this work to the paper by Kogar et al., leading the authors to an interpretation of the data in terms of spontaneous symmetry-breaking

scenario. The scenario of Kogar et al., where the checkerboard modulation is nucleated around (photoinduced) topological defects, seems, however, much simpler. In principle, photoinduced changes in the Fermi-surface topology, together with the phonon branch/momentum dependent electron-phonon coupling can indeed be the underlying origin of the instability. But a proper motivation and arguments for this scenario are missing. I could imagine that such scenario could be motivated based on the threshold energy being smaller than the thermal energy to drive the phase transition. On the other hand, I do not see the observation being inconsistent with the scenario put forward by Kogar et al.

I also cannot follow the arguments of the soft-mode-driven scenario. Per definition, the soft modes are modes above the critical temperature, at the wavevector close to q_a , which freeze when the static modulation appears at T_c . On page 14, the authors argue to be able to trace the soft phonon dynamics by following the dynamics of different lattice reflections. If I am not mistaken, the intensity of the lattice reflection is given by temperature (conventional Debye-Waller effect) and the periodic lattice distortion (as schematically first shown in [20]). So, by suppressing the periodic lattice distortion along the c-axis, the [600] lattice reflection gains strength due to the nature of the distortion (transverse phonons get frozen when the lattice modulation along the c-axis takes place in equilibrium), while [006] reflection experiences only a drop in intensity. Qualitatively, this seems perfectly consistent with the experimental observations reported here (quantitatively, things are likely very complicated, especially given the fact that the system is likely highly inhomogeneous in the excited state). But I do not understand how the authors extract the dynamics of soft modes (modes at finite frequency and momentum) from these data.

Furthermore, I still do not understand the analysis of the data with three temperature model. Why is this relevant? At short times, clearly the description with temperatures should not be adequate, especially not in systems displaying ordering, signified by gaps periodic modulation etc. Here a phenomenological model like the time-dependent Ginzburg-Landau model makes much more sense. What do we actually learn from the TTM analysis?

Reviewer #2 (Remarks to the Author):

After reading the revised manuscript and the reply of authors, I feel that the authors have answered all my questions. I recommend the revised manuscript for publication in Nature communications.

Also, I must admit that the paper by Kogar et al., mentioned by the first referee, had previously skipped my attention. I found it interesting to see the two works next to each other. I agree with the reply of the authors of the present manuscript that that these two works, although on similar systems and with similar observations, are sufficiently distinct. After all, these works apparently have appeared on arXiv more or less at the same time.

Summary of Changes

We thank both Reviewers for their thoughtful comments on our manuscript. While Reviewer 2 is satisfied with our revision, Reviewer 1 raises several questions for clarification. We acknowledge these issues and an effort to properly address them (see detailed replies below). One main concern is on comparing our work with the paper on LaTe_3 by Kogar et al (Ref. 41), which shows similar a -CDW formation. The authors interpreted the phenomenon as a transient order developed by nucleating around light-induced topological defects. Our argument, centered on non-equilibrium processes and spontaneous symmetry breaking (SSB), represents a very different interpretation. We do wish to point out that our results contain important new findings (as explicitly outlined below) supporting this scenario. Given there are two different interpretations on the phenomenon, we fully agree with Reviewer 1 that a critical assessment of the results and interpretations should be given in our paper.

To address Reviewer 1's critiques, two main changes are made in the revised manuscript, outlined here.

- For clarification purpose (to address the questions 2-4), we have provided theoretical framework of the scattering from non-equilibrium system involving dynamical responses of lattice phonons and CDW state. Accordingly, we redefine the notation for the phonons (with the coordinate \mathbf{q}) and the collective modes of the CDW state (i.e. fluctuations waves of the static CDW order with the coordinate \mathbf{k}). The corresponding ordering vectors for the main lattice (\mathbf{G}) and the CDW sub-lattice (\mathbf{Q}) are capitalized. The scattering structure factors from these different contributions, S_G for the lattice and S_Q for the CDW, are derived accordingly; see Methods Sec. 3.

Landau-Ginzburg model for non-thermal and thermal CDW phase transitions | The two-dimensional free energy landscape, $f(\eta_c, \eta_a)$ where η_c and η_a are order parameters of the c - and a -CDW states, for photoinduced phase transition modelled after the experiment condition at $F=1.85 \text{ mJ/cm}^2$. The non-equilibrium pathway (depicted in the left route) involves the formation of a bi-directional hidden state when the repulsive coupling potential is suppressed by a swift reduction of the preeminent order parameter $|\eta_c|$. In contrast, in the thermal pathway (right) the global free energy minimum remains in the condition where $|\eta_a| = 0$, namely the CDW is uniaxial.

- One most important new component (to address Question 1) is the introduction of a Landau-Ginzburg (L-G) theory with a (repulsive) bi-quadratically coupled interaction potential to describe the competition between c - and a -CDWs in this system. This model (see Fig. 2a and reproduced above for free energies at different stages of evolution) is introduced on page 6 and elaborated in the Method Sec.2. Specifically, the phenomenological L-G framework embodies the essential non-equilibrium physics for the formation of hidden (bi-directional) states; its introduction can tacitly capture the key findings of our experiments, which are listed below.
 - (1) **Non-equilibrium interaction quench and SSB:** The competitively coupled L-G model bears on a hidden bidirectional instability approaching the thermal critical point in the $R\text{Te}_3$ system (Ref. 23 and discussed on page 6). We attribute the sudden suppression of the preeminent c -CDW by laser pulses as the non-thermal interaction quench that causes a reduction of the repulsive coupling interaction. The interaction quench enhances the relative strength of a -CDW and leads to a new CDW formation with bi-directional components. The interaction quench scenario is discussed along with the L-G model on page 6.
 - (2) **A slow onset of the new phase.** As illustrated in the stage II of the figure, the occurrence of the new phase proceeds following the swift reduction of the c -CDW order parameter, $|\eta_c|$. This L-G picture of a serial process for forming a new bi-directionally ordered state naturally describes the slow onset of a -CDW observed in our experiment. More subtleties related to the non-equilibrium scattering problems specific to CDW formation are given in Fig. 2b and discussed on pages 7, 8 and 10. Briefly, to form the long-range parameter, the process involves paring of the soft modes into collective excitations and coarsening – a relatively slow dynamics compared to melting especially given the new 2D landscape is significantly softened at this stage. The result is discussed on page 8.
 - (3) **Coupled a - and c_+ -CDW dynamics:** Due to the competitive nature, introducing the a -CDW into the transformed c -CDW order can drive the reorganization of the system. This reorganization is featured in the emergence of a c^+ -component, with a transformed larger ordering vector Q_{c^+} (≈ 0.30 r.l.u.) than the initial state Q_c (≈ 0.28 r.l.u.). Here, the changes are tied to the Fermi Surface topology, where a larger ordering vector reflects a reduced curvature in the Fermi Surface contour – an established feature known from APRES in the heavier $R\text{Te}_3$ members as they adopt a bi-directional ordering (typically with $Q_c = 0.30\text{-}0.32$ r.l.u. in contrast to the 0.28 r.l.u. of the uniaxial state); see discussions on page 11 and page 16. The co-coarsening of a - and c^+ -CDW components with increased Q_a and Q_{c^+} to ≈ 0.30 r.l.u. are observed, indicating an evolution into the bidirectional state. The results are discussed on pages 8 and 11.
 - (4) **Threshold behavior for introducing a -CDW order:** According to the L-G theory, to tilt the symmetry-breaking landscape in favor of the bi-directional state a *minimum suppression* of the c -CDW is required; see page 6. This threshold behavior (as identified by $F_{c,a}$) is indeed observed in our experiment – see Fig. 3d and discussed on pages 10, and 20.
 - (5) **Involvement of soft modes:** Since the emergence of soft modes is a hallmark of continuous phase transition, we cite our observation of soft-mode excitations upon laser quench along both symmetry-breaking axes as the evidences that the non-equilibrium SSB (prescribed by the L-G model) is at play. Importantly, the non-adiabaticity evidenced in the first excitation of incoherent soft modes, followed by collective mode dynamics forming the new CDW state is a key signature of SSB driven by a quench. This topic is discussed on page 15 and Fig. 5d.

We now address specific questions raised by the reviewers.

Reviewer 1:

1) In their rebuttal and the revised version of the manuscript the authors improved both the arguments as well as presentation. However, I still have problems connecting the proposed interpretation with the presented data (see below).

First, I apologize for not realizing that the first version of this publication has been posted on arxiv even before the post (arXiv:1904.07472v1) of a recently published work by Kogar et al. (Nature Physics 2019, <https://doi.org/10.1038/s41567-019-0705-3>). Given the fact that the two works present (in my opinion) nearly the same result, I can understand the authors' frustration when receiving my report.

Apparently, the publication of "Nonequilibrium dynamics of spontaneous symmetry breaking into a hidden state of charge-density wave" has already been declined elsewhere, before submitting it to Nature Communications. Perhaps one of the reasons for declining publication was the "novelty" argument in view of the similar work by Kogar et al. (which would be unfortunate). On the other hand, I can imagine the reason being the mentioned lack of support for the proposed scenario. Moreover, instead on using the published results and the interpretation as a starting point (pointing out the additional information that can be obtained from the data), the authors simply make a remark that similar results have been published, without critical assessing the data and the proposed model. In my opinion, this is contra productive.

Our reply: We thanks the Reviewer for the kind understanding and have implemented the changes recommended by the Reviewer. First, we have listed findings by Kogar et al. on LaTe_3 and discussed their interpretation based on the nucleation around light-induced topological defects, and recent related literatures on this topic; see the second paragraph on page 4. Our arguments have been based on the non-equilibrium symmetry breaking, which is a rather different interpretation of the results. As discussed in the Summary of Changes, we have made a more tacit introduction of this concept using the Landau-Ginzburg theory that, we believe, captures all essential features of our experiments in a natural way. We do want to point out that, while there are similarities between our results, there are notable key differences. It should be fair to state that we were able to achieve the fs scattering with a higher momentum solution, taking advantage of a setup that probes a much smaller sample ($30 \mu\text{m}$ vs $>100 \mu\text{m}$) at a lower beam energy (100 keV vs MeV) under a higher repetition rate (1 kHz vs. 120 Hz). This new arrangement gives us more precision to pinpoint the non-equilibrium dynamics important to unveil the fundamentally competitive symmetry-breaking nature of the phase transitions.

2) I am turning now to the revised manuscript at hand. The work reads well up to the page 8, when the main experimental result is being discussed (in reference to Figure 2 on page 7). Here I get stuck and frustrated. In my opinion, this part presents the main problem of the manuscript. If I understand correctly, the support for the proposed scenario should be given by Figure 2c, showing the time evolution of the diffraction intensity profiles in the vicinity of the CDW wavevectors q_c and q_a . First, it would be easier for the reader to have the profiles along the c^* and a^* directions plotted separately. Moreover, it would be nice to note that traces in Fig. 2a have different signs. However, my main problem lies in the claimed appearance of a new (transient) order along c^* at the wavevector c_+ different from the equilibrium one (q_c). On page 8 it is argued that $q_{c_+} = q_c + 0.07 c^*$, i.e. $q_c = 0.28 c^*$, while $q_{c_+} = 0.35 c^*$. This follows from the definition of $k = q - q_c$ on page 8, as well as from the arrow in the upper panel of Figure 2c. I simply do not see any evidence supporting this claim in the raw data (neither in the main text, nor in the supplementary). On page 12, however, the authors quote $q_{c_+} = 0.29 c^*$, which is more along the lines with the data in Panel 2c and can barely be distinguished from $q_c = 0.28 c^*$. The authors argue that the distinction between the two orders/periodicities (at q_c and q_{c_+}) can be made from the analysis of the superlattice peak with two Voigt functions. All I can see here – in the first approximation – is a broadening of the linewidth. But I see evidence in the data for a major change in periodicity along the c -axis direction (i.e. $q_{c_+} = 0.35 c^*$). Perhaps a two Voigt function fit is indeed better than a single Voigt function, but this should not be much of a surprise as the system after excitation is likely inhomogeneous. The simple reason could be that the

excitation profile as such is inhomogeneous, since the optical penetration depth (20 nm) is comparable to the sample thickness (25 nm). (Note that in the work by Kogar et al. thicker samples are used, making the excitation even more inhomogeneous. This is manifested by weaker suppression of the CDW peak at q_c at higher excitation densities). Anyway, since both referees pointed out the issue of q_{c+} vs. q_c in original reports, I would have expected the authors to address it.

Our reply: After carefully reading through the Reviewer’s comments, we believe there is a misunderstanding about the meaning of “ k ” used in our paper, which leads to the inconsistency cited by the Reviewer. Our definition of $k = q - q_c$ on page 8 of the original text is meant to address the *diffuse scattering* amplitude (here k is referred to as the momentum coordinate in the CDW sublattice; in contrast to q which is the momentum coordinate of the main lattice). We also do want to point out the label in Fig. 3a for the scattering at $k=0.07$ r.l.u. was misplaced in the diffraction pattern and has been corrected. The purpose of citing scattering at $k=0.07$ r.l.u. simply is to show the different dynamics in the diffuse component away from the central peak, which in fact is predominantly contributed by the emerging c^+ -component in the formation of a bi-directional state. We further point out the new component differs also in the magnitude of the ordering vector, which is closer to 0.30 r.l.u. The argument about uniaxial to bi-directional state transition driven by the non-thermal quench is given by the L-G model summarized above. Discussion on this topic is given in the paper on page 11 and page 16.

In order to avoid such confusion in the notation, we have changed the convention, using capital Q in addressing the CDW momentum wavevector, and reserving q and k for the momentum coordinates of the main lattice and CDW-sublattice. The same convention is used throughout the text and the Methods sections.

On the possibility of effects from excitation inhomogeneity, we believe this is indeed an issue to consider especially on a thicker sample, but not significantly enough in our experiment to change our interpretation of the results. It is worth noting that in a thin film geometry, the optical absorption no longer observes the exponential decay as in the bulk system. This is due to the optical interferences in the thin film geometry. This effect can be accounted by solving the Maxwell equation with boundary conditions; see e.g. Ref. 29. In this case, the multiply reflected waves from the back (and front) surfaces of the film must be considered. The interferences typically result in a flattened absorption curve, especially when the film thickness is close to the inherent decay length of a bulk sample – a scenario applies to our experiment. Our calculation (included as Extended Data Fig. 1 and discussed in Methods Sec. 1) shows that the depth(z) inhomogeneity of the laser field to be on the level of $\approx \pm 10\%$. This leads to a variation in the excitation energy density across the entire film of $\approx \pm 20\%$.

3) The way I understand it, assuming a new periodicity of the new order with changed periodicity along the c -axis direction is the most important (experimental) distinction of this work to the paper by Kogar et al., leading the authors to an interpretation of the data in terms of spontaneous symmetry-breaking scenario. The scenario of Kogar et al., where the checkerboard modulation is nucleated around (photoinduced) topological defects, seems, however, much simpler. In principle, photoinduced changes in the Fermi-surface topology, together with the phonon branch/momentum dependent electron-phonon coupling can indeed be the underlying origin of the instability. But a proper motivation and arguments for this scenario are missing. I could imagine that such scenario could be motivated based on the threshold energy being smaller than the thermal energy to drive the phase transition. On the other hand, I do not see the observation being inconsistent with the scenario put forward by Kogar et al.

Our reply: We have addressed the different interpretation issue and related argument in our reply to question (1). We strongly argue that the observation of a higher momentum wavevector and the threshold

behavior, as also indicated by the Reviewer, along with other non-equilibrium features outlined in the Summary of Changes are supportive of the non-equilibrium SSB scenario, captured by the L-G model. We have taken on the Reviewer 1's suggestions and made corresponding changes across the revised manuscript, as outlined in the Summary of Changes.

4) I also cannot follow the arguments of the soft-mode-driven scenario. Per definition, the soft modes are modes above the critical temperature, at the wavevector close to q_a , which freeze when the static modulation appears at T_c . On page 14, the authors argue to be able to trace the soft phonon dynamics by following the dynamics of different lattice reflections. If I am not mistaken, the intensity of the lattice reflection is given by temperature (conventional Debye-Waller effect) and the periodic lattice distortion (as schematically first shown in [20]). So, by suppressing the periodic lattice distortion along the c-axis, the [600] lattice reflection gains strength due to the nature of the distortion (transverse phonons get frozen when the lattice modulation along the c-axis takes place in equilibrium), while [006] reflection experiences only a drop in intensity. Qualitatively, this seems perfectly consistent with the experimental observations reported here (quantitatively, things are likely very complicated, especially given the fact that the system is likely highly inhomogeneous in the excited state). But I do not understand how the authors extract the dynamics of soft modes (modes at finite frequency and momentum) from these data.

Our reply: To give proper answers to address Reviewer's question, we give more information on the topic of Debye-Waller Factor (DWF) analysis; see Methods Sec. 3. Our definition follows the convention used in the literature, which separates the DWF from the contribution originated in the symmetry-breaking or recovery. In this scheme, then the effect the Reviewer referred to is about the symmetry recovery term, which is separated from the decay factor associated with the DWF. To better see how these two different effects impact the intensities of the lattice and CDW peaks, we give the derivations for the respective structure factors starting from the general expression of the lattice displacement $\mathbf{u}_L = \mathbf{u}_q + \mathbf{u}_\eta$, separating the two different origins. We show the phonon term, \mathbf{u}_q , lead to the typical DWF, $e^{-2M_{hkl}(t)}$. Meanwhile, the \mathbf{u}_η representing the static distortion from the CDW leads to an independent term described by the Bessel function. The two different factors can be independently extracted from analyzing the lattice and CDW structure factor evolutions as outlined in Methods Sec. 3.

We also offer the reasoning on how one differentiates the soft modes, among all phonons in the system, as described on page 13 and page 15, in discussing the DWF. Normally, the phonons in a symmetric 2D lattice give an isotropic vibrational profile probed through the DWF at different Bragg peak positions, \mathbf{G}_{hkl} . The introduction of CDW breaks the translational symmetry along a specific ordering vector which leads to strong anisotropy in the vibrational profile. The modes excited, mainly the soft modes, approaching the phase transitions, are characterized by the highly anisotropic \cos^2 -distributions, which allow one to see the emergence of soft modes in the vibration polar plot, as outlines in Fig. 4c.

On page 15, we also give an explanation on how one may trace the kinetic energy redistribution in the non-equilibrium system through the evolution of the vibrational profiles. The two soft-mode branches connected to c- and a-CDWs are isolated by the vibration polar plots in Fig. 5c. Clearly shown are the modes along the two symmetry-breaking axes having very different amplitude changes. As in the conventional interpretation of the DWF, these amplitudes inform the kinetic energy in the manifestation of lattice vibrations – but following equipartition in an equilibrium system. In contrast, for a non-equilibrium system they are representative of the *local effective temperatures* in isolated sub-systems. Hence, the thermalization of the two systems can be studied by how such vibration polar profile becomes more isotropic – signifying the energy spreads into other 2D phonons not initially excited. The discussion of the relevant results (Figs. 5c&d) is given on page 15.

5) Furthermore, I still do not understand the analysis of the data with three temperature model. Why is this relevant? At short times, clearly the description with temperatures should not be adequate, especially not in systems displaying ordering, signified by gaps periodic modulation etc. Here a phenomenological model like the time-dependent Ginzburg-Landau model makes much more sense. What do we actually learn from the TTM analysis?

Our reply: The three-temperature model (TTM) focused on the microscopic evolution, and was meant to address the question on the transient non-equilibrium state where the fluence-dependent recovery dynamics, as characterized by the τ_e in Fig. 3b, is related to the thermalization of the system. However, we understand introducing this model actually gave unnecessary complications and may distract the reader. In the revised manuscript, the non-equilibrium scenario itself is self-evident in the different soft-mode dynamics connected to the two CDW systems – see the discussion above. We instead focus the argument of thermalization process based on the experiment results and remove the TTM from our manuscript.

Reviewer 2:

After reading the revised manuscript and the reply of authors, I feel that the authors have answered all my questions. I recommend the revised manuscript for publication in Nature communications.

Also, I must admit that the paper by Kogar et al., mentioned by the first referee, had previously skipped my attention. I found it interesting to see the two works next to each other. I agree with the reply of the authors of the present manuscript that that these two works, although on similar systems and with similar observations, are sufficiently distinct. After all, these works apparently have appeared on arXiv more or less at the same time.

Our reply: We thank very much the Reviewer 2's encouraging comments and the helpful remarks in the previous review that continue to shape this revision.